# An Unconditional Representation of the Conditional Score in Infinite-Dimensional Linear Inverse Problems

**Fabian Schneider**                                    *fabian.schneider@lut.fi*
*School of Engineering Science*
*Lappeenranta-Lahti University of Technology*
*Vienna University of Technology (TU Wien)*

**Duc-Lam Duong**                                       *duc-lam.duong@lut.fi*
*School of Engineering Science*
*Lappeenranta-Lahti University of Technology*

**Matti Lassas**                                        *matti.lassas@helsinki.fi*
*Department of Mathematics and Statistics*
*University of Helsinki*

**Maarten V. de Hoop**                                  *mdehoop@rice.edu*
*Department of Computational and Applied Mathematics*
*Rice University*

**Tapio Helin**                                         *tapio.helin@lut.fi*
*School of Engineering Science*
*Lappeenranta-Lahti University of Technology*

**Reviewed on OpenReview:** *https://openreview.net/forum?id=rO8erhXHPo*

## Abstract

Score-based diffusion models (SDMs) have emerged as a powerful tool for sampling from the posterior distribution in Bayesian inverse problems. However, existing methods often require multiple evaluations of the forward mapping to generate a single sample, resulting in significant computational costs for large-scale inverse problems. To address this, we propose an unconditional representation of the conditional score function (UCoS) tailored to linear inverse problems, which avoids forward model evaluations during sampling by shifting computational effort to an offline training phase. In this phase, a *task-dependent* score function is learned based on the linear forward operator. Crucially, we show that the conditional score can be derived *exactly* from a trained (unconditional) score using affine transformations, eliminating the need for conditional score approximations. Our approach is formulated in infinite-dimensional function spaces, making it inherently discretization-invariant. We support this formulation with a rigorous convergence analysis that justifies UCoS beyond any specific discretization. Finally we validate UCoS through high-dimensional computed tomography (CT) and image deblurring experiments, demonstrating both scalability and accuracy.

## 1 Introduction

Inverse problems seek to determine unknown quantities through indirect and noisy measurements, typically leading to ill-posed scenarios. The Bayesian approach to inverse problems frames the task as a quest for information. Blending statistical prior information of the unknown with a likelihood model for the measurement data gives rise to a posterior distribution, which fully characterizes the unknown conditioned on noisy data Kaipio & Somersalo (2006); Stuart (2010). In severely ill-posed problems, the quality of inference

is strongly dependent on the expressivity of the prior. Traditional hand-crafted priors, such as the total-variation prior, tend not to be expressive enough to characterize complicated structures Sun et al. (2023). Generative models offer a flexible and computationally feasible approach to prior modeling as they offer the possibility of generating new samples after training on a data set characterizing the prior.

This work investigates sampling from the posterior distribution of linear inverse problems using score-based diffusion models (SDMs) Song et al. (2021), which have recently received wide attention in the literature (in the context of inverse problems, see e.g. Batzolis et al. (2021); Lim et al. (2025); Hagemann et al. (2025); Graikos et al. (2022); Feng et al. (2023); Sun et al. (2023); Pidstrigach et al. (2024); Dey et al. (2024); Holzschuh et al. (2023); Dou & Song (2023); Barbano et al. (2025); Cardoso et al. (2023); Feng & Bouman (2023); Song et al. (2024); Meng & Kabashima (2022); Kveton et al. (2024); Wu et al. (2024a;b); Baldassari et al. (2024b); Yao et al. (2025); Chen et al. (2025)). An SDM consists of two main components: a forward diffusion process and a reverse generative process. In the forward diffusion, the model gradually transforms the target distribution into a simpler, tractable distribution with a specific stochastic differential equation (SDE) driven by Gaussian noise. The generative process simulates the time-reversal of the diffusion process with a backwards SDE. This denoising phase relies on the drift, which is computed using the logarithmic gradients (scores) of the diffused data densities. These scores are typically estimated using a neural network, allowing efficient simulation of the backwards SDE and, consequently, sample generation from the target distribution. SDMs have demonstrated significant success across a variety of domains, including inverse problems such as medical imaging Dey et al. (2024); Levac et al. (2023); Barbano et al. (2025); Chung et al. (2023b); Song et al. (2022).

For inverse problems, posterior sampling with SDMs involves the challenging task of estimating the score function conditioned on the measurements. A key difficulty is balancing computational efficiency with scalability while ensuring accurate posterior sampling. Motivated by the limitations of existing approaches, we propose a novel method for conditional sampling, based on an *unconditional task-dependent representation of the conditional score (UCoS)* that overcomes these challenges by leveraging an explicit likelihood model, which is often available and predefined in inverse problems. UCoS is particularly well-suited for large-scale imaging tasks that require efficient posterior sampling for varying sets of measurement data.

The existing posterior sampling methods can be broadly categorized into three approaches. The *first* approach modifies the unconditional reverse diffusion process to guide sample trajectories towards the posterior, either by adding a correction term to the score Jalal et al. (2021); Chung et al. (2023a); Song et al. (2023); Adam et al. (2022); Chung & Ye (2022); Levac et al. (2023) or incorporating a data consistency optimization step Song et al. (2022); Dey et al. (2024); Chung et al. (2022). The *second* approach employs gradient-based Monte Carlo sampling techniques, replacing the prior score with a learned score Cardoso et al. (2023); Sun et al. (2023); Feng et al. (2023), with recent extensions to infinite dimensions Baldassari et al. (2024b). For a recent review of methods largely following these two approaches (in finite dimensions), we refer to Daras et al. (2024). A key advantage of these approaches is that they allow the use of a pre-trained unconditional prior score, accommodating different imaging tasks simultaneously without requiring task-specific training. This is particularly useful for handling extensive multipurpose training sets in applications such as image processing. However, such methods suffer from (1) high computational costs due to repeated forward evaluations, (2) poor scalability in high-dimensional or large-scale inverse problems, and (3) potential inconsistency in posterior samples due to the lack of rigorous convergence guarantees.

The *third* approach directly trains a *conditional* score function to perform conditional sampling, learning an amortized version of the conditional score that depends on observations Baldassari et al. (2024a); Batzolis et al. (2021). This method extends rigorously to infinite-dimensional diffusion models, making it attractive for large-scale inverse problems, as it eliminates the need to evaluate the forward map during sampling. However, it requires extensive training data on the joint distribution to reliably approximate the conditional score function, particularly as the problem dimensionality increases.

We address the prohibitive sampling cost for large-scale inverse problems by demonstrating that this computational overhead can be offloaded to the training phase, thereby improving the efficiency of posterior sampling. This is achieved by designing a *task-dependent* training phase that depends on the forward mapping but *not* on the measurement data. More precisely, we introduce a diffusion-like random process whose

distribution is explicitly dependent on the forward mapping, allowing the score (which we term the *task-dependent score*) to be estimated using standard methods. We then demonstrate that the conditional score corresponding to the posterior distribution can be recovered from this task-dependent *unconditional* score through simple affine transformations involving the measurement data. Furthermore, we modify the training procedure to enable evaluation of the conditional score without requiring application of the forward mapping during sampling. In comparison to the conditional method, integrating the explicit likelihood model into the process reduces the dimensionality of the training target. As a result, UCoS offers the best of both worlds: like the conditional approach, it avoids forward mapping evaluations during posterior sampling while its score model remains as lightweight as standard unconditional ones. This leads to more favorable scaling of the online sampling cost as the size or dimensionality of the problem grows.

Inverse problems are often about inferring quantities represented by functions such as solutions or parameters to PDEs or integral equations. These models must be arbitrarily discretized to obtain a finite-dimensional computational model Mueller & Siltanen (2012). In Bayesian inversion, there has been a long-standing effort to develop methods that are discretization-independent Lehtinen et al. (1989); Lassas & Siltanen (2004); Lassas et al. (2009), aligned with the principle to "avoid discretization until the last possible moment" Stuart (2010). This is also critical for the success of SDMs in Bayesian inversion as recent theoretical studies indicate that the performance guarantees do not always generalize well on increasing dimension Chen et al. (2023b); Bortoli (2022); Pidstrigach et al. (2024). Inspired by recent developments on defining the score-based diffusion framework in infinite-dimensional spaces Pidstrigach et al. (2024); Baldassari et al. (2024a), we also extend our method rigorously to a separable Hilbert space setting. In particular, in the spirit of Pidstrigach et al. (2024), we perform a convergence analysis of UCoS, establishing a bound between the samples generated by UCoS and the true posterior target measure. Moreover, we conduct numerical experiments of inverse problems related to computerized tomography (CT) and deblurring to illustrate the practical applicability of UCoS.

## 1.1 Related work

The body of literature on SDMs is growing rapidly. Let us mention that Song et al. (2021) developed a unified framework combining score-based Hyvärinen & Dayan (2005); Song & Ermon (2019) and diffusion Sohl-Dickstein et al. (2015); Ho et al. (2020) models to interpret SDMs as a time-reversal of certain SDEs. Our paper is inspired by the work on conditional SDMs in the context of inverse problems. One line of work seeks to modify the unconditional prior score function to generate samples that approximately follow the posterior distribution. Examples of this include projection-based approaches Song et al. (2022); Dey et al. (2024), which project the samples on a subspace that solves the inverse problem during the generating process; methods based on the gradient of a log-likelihood Jalal et al. (2021); Chung et al. (2023a); Levac et al. (2023) and plug and play approaches Graikos et al. (2022) which add appropriately chosen constraint-terms to the unconditional score function to steer the process towards generating desirable samples; and the Doob's *h*-transform approach Denker et al. (2024) which learns a small, auxiliary correction to the unconditional score function. A closely related approach is the Monte Carlo guided diffusion framework (MCGDiff) Cardoso et al. (2023), which leverages posterior samples from a linear Bayesian inverse problem obtained via sequential Monte Carlo methods using the singular value decomposition (SVD) to guide the training of a diffusion model. The theoretical foundation of MCGDiff however builds upon the availability of the full SVD of the forward operator, which restricts its applicability to mildly ill-posed problems and moderate-dimensional settings where the SVD can be computed and inverted stably. In contrast, UCoS is designed to operate in a fully *matrix-free* manner, requiring access only to the forward map and its adjoint.

Another line of work seeks to approximate the conditional score function of the posterior distribution directly Saharia et al. (2023); Batzolis et al. (2021); Baldassari et al. (2024a). As this approach increases the input dimension of the score function drastically, the training process is more computationally expensive and requires more high quality training samples, which may be restrictive, especially in very high-dimensional problems.

The theory of infinite-dimensional SDMs has been initiated only very recently. Hagemann et al. (2025) modifies the training phase of diffusion models to enable simultaneous training on multiple discretization levels of functions and proves the consistency of their method. Lim et al. (2025) also generalizes the trained

model over multiple discretization levels proposing to generate samples with the annealed Langevin algorithm in infinite dimensions. Pidstrigach et al. (2024) was the first to formulate the SDM directly in infinite-dimensional space, demonstrating that the formulation is well-posed and providing theoretical guarantees. Our work is closely connected to Baldassari et al. (2024a), where the authors introduce the conditional score in an infinite-dimensional setting. Moreover, they provide a set of conditions to be satisfied to ensure robustness of the generative method and prove that the conditional score can be estimated via a conditional denoising score matching objective in infinite dimensions.

## 1.2 Main contribution

This work presents a novel and scalable framework for posterior sampling in high-dimensional and infinite-dimensional inverse problems. The key insight is that posterior samples generated via conditional SDMs can be computed without any evaluation of the forward operator during sampling. Instead, without introducing any errors, the computational effort can be shifted to the offline task of training the *unconditional* score of a specific diffusion-like random process. This foundational principle generalizes to other infinite-dimensional diffusion models, beyond the Ornstein-Uhlenbeck process studied here. More precisely,

- In Theorem 3.7 we establish an identity for the conditional score connecting it to a task-dependent unconditional score through affine transformations depending on the forward mapping and the measurement data. The theorem extends this general principle to an infinite-dimensional setting.

- In Theorem 4.1 we derive an error estimate for the generative process following the ideas suggested by Pidstrigach et al. (2024). In particular, the error estimate explicitly underscores the contribution of the loss function employed in training.

We numerically explore our method in Section 5, demonstrating that task-dependent training is effective in practice and that online posterior sampling can be performed without evaluations of the forward operator.

## 2 Background

### 2.1 Score-based diffusion models in infinite dimensions

**Finite-dimensional score-based diffusion models**  Score-based diffusion models (SDMs) are state-of-the-art machine learning generative models (Song et al. (2021)) that learn a data distribution through gradual denoising of a normally distributed random variable. A diffusion process diffuses an image $X_0 \sim p_0$ from $t = 0$ to $t = T$ via the SDE (Ornstein-Uhlenbeck process, or OU)

$$\mathrm{d}X_t = -\frac{1}{2}X_t\mathrm{d}t + \mathrm{d}W_t,$$

where the marginal densities are denoted by $p_t$, in particular, for large $T$, $p_T$ is close to the Gaussian density $\mathcal{N}(0, I)$. The marginal densities are then reversed using the backward SDE

$$\mathrm{d}Y_t = \frac{1}{2}Y_t\mathrm{d}t + \nabla_x \log p_{T-t}(Y_t)\mathrm{d}t + \mathrm{d}W_t,$$

such that $Y_T \sim p_0$.

**Infinite-dimensional score-based diffusion models**  Let us now review the unconditional SDMs in infinite dimensions proposed by Pidstrigach et al. (2024). Let $\mu$ be the target distribution, supported on a separable Hilbert space $(H, \langle \cdot, \cdot \rangle_H)$. Let $\{X_t^\mu\}_{t=0}^T$ stand for the infinite-dimensional diffusion process for a continuous time variable $t \in [0, T]$ satisfying the following infinite-dimensional SDE

$$dX_t^\mu = -\frac{1}{2}X_t^\mu dt + \mathcal{C}^{1/2}dB_t, \tag{1}$$

where $\mathcal{C}$ is a fixed trace class, positive-definite, symmetric covariance operator $\mathcal{C} : H \to H$ and $B_t$ is a Wiener process on $H$ with covariance $tI$, see Appendix A.2. We assume that the process $X_t^\mu$ is initialized with $\mu$,

i.e., $X^\mu \sim \mu$. Notice carefully that we embed the initial condition $\mu$ to the notation $X_t^\mu$, $t \geq 0$, as we later analyze the interplay between different initializations. Here and in what follows, we assume that the initial conditions and the driving process are independent. We adopt the formal definition of the score functions by Pidstrigach et al. (2024):

**Definition 2.1.** Define the (weighted, unconditional) score function $s(x, t; \mu)$ for $x \in H$ as

$$s(x, t; \mu) = -\frac{1}{1 - e^{-t}} \left( x - e^{-t/2} \mathbb{E}(X_0^\mu | X_t^\mu = x) \right). \tag{2}$$

*Remark* 2.2. Assume that $H = \mathbb{R}^n$ and that the distribution $\mu$ admits a density $p_0$ with respect to the Lebesgue measure. For any $t \geq 0$ the random variable $X_t^\mu$ obtained through SDE in Equation 1 has a density $p_t$. The unconditional score $s(x, t; \mu)$ given in Equation 2 then satisfies $s(x, t; \mu) = \mathcal{C} \nabla_x \log p_t(x)$. In other words, in finite dimensions, Definition 2.1 reduces to the common score identified as the log-gradient of the density, scaled by the covariance matrix $\mathcal{C} \in \mathbb{R}^{n \times n}$.

Now assume that for $T > 0$,

$$\sup_{t \in [0,T]} \mathbb{E} \|s(X_t^\mu, t; \mu)\|_H^2 < \infty, \tag{3}$$

It is shown in Pidstrigach et al. (2024) that given $Y_T^\mu \sim \mathscr{L}(X_T^\mu)$ the following SDE

$$dY_t^\mu = \left( -\frac{1}{2} Y_t^\mu - s(Y_t^\mu, t; \mu) \right) dt + \mathcal{C}^{1/2} d\bar{B}_t \tag{4}$$

is the time-reversal of Equation 1, where $\bar{B}_t$ is a different Wiener process on $H$. The process $Y_t^\mu$ is independent of the past increments of $\bar{B}_t$, but not of the future ones, see Anderson (1982).

**Training and sampling** In both finite- and infinite-dimensional diffusion models, the score function is learned by a neural network $s_\theta(\cdot, \cdot; \mu) : H \times [0, t] \to H$ such that $s_\theta(x, t; \mu) \approx s(x, t; \mu)$ (where in finite dimension $H = \mathbb{R}^n$ we interpret $s(x, t; \mu)$ as $\nabla \log p(x)$). A common technique to enable empirical learning of the score function is conditional denoising score matching, introduced in Vincent (2011). This method has been shown to work in the conditional Batzolis et al. (2021) and infinite-dimensional setting Baldassari et al. (2024a) for the forward SDE given by the OU process. Training is performed via stochastic gradient descent by minimizing a score-matching objective (loss function),

$$\text{SM}(s_\theta) = \mathbb{E}_{t \sim U[\delta, T]} \mathbb{E}_{x \sim \mathscr{L}(X_t^\mu)} \lambda(t)^2 \|s_\theta(x, t; \mu) - s(x, t; \mu)\|_H^2, \tag{5}$$

over some appropriate class of neural network. Here $\lambda : [\delta, T] \to \mathbb{R}^+$ is a positive weighted function and $0 \leq \delta < T$ a truncation that avoids numerical instability for small times. In general, the score function $s$ is intractable, such that a denoising score-matching objective is introduced,

$$\text{DSM}(s_\theta) = \mathbb{E}_{t \sim U[\delta, T]} \mathbb{E}_{(x, x_0) \sim \mathscr{L}(X_t^\mu, X_0^\mu)} \lambda(t)^2 \|s_\theta(x, t; \mu) - (1 - e^{-t})^{-\frac{1}{2}} (x - e^{-\frac{t}{2}} x_0)\|_H^2.$$

It can be shown, that the two objectives equal up to a constant that does not depend on the parameter $\theta$. After training a score $s_\theta(\cdot, t)$ for all time $t \in [0, T]$, we can then generate samples from $\mu$ by running the backward SDE Equation 4 with the trained score $s_\theta$ instead of $s$. The solution of the backward SDE can then be numerically solved using traditional methods such as Euler–Maruyama approximations.

## 2.2 Bayesian inverse problems and conditional score

In this paper, we focus on the setting, where the score corresponding to the reverse drift is conditioned on observations. We consider a linear inverse problem

$$y = Ax + \epsilon, \tag{6}$$

where $x \in H$ is the unknown in some separable Hilbert space $H$, $y \in \mathbb{R}^m$ stands for the measurement and $A : H \to \mathbb{R}^m$ is a bounded linear operator. The random noise is modeled by a centered Gaussian

distribution $\epsilon \sim \mathcal{N}(0, \Gamma)$, $\Gamma \in \mathbb{R}^{m \times m}$. We adopt a Bayesian approach to inverse problems Stuart (2010) and assume to have some prior knowledge of the distribution of $x$ before any measurement is made. This knowledge is encoded in a given prior $\mu$, defined as a probability measure on $H$. This approach gives rise to a posterior that is absolutely continuous with respect to the prior and its Radon–Nikodym derivative is given by, $\mu$-almost everywhere,

$$
\begin{aligned}
\frac{\mathrm{d}\mu^y}{\mathrm{d}\mu}(x) &= \frac{1}{Z(y)} \exp\left(-\frac{1}{2}\|Ax - y\|_\Gamma^2\right), \\
Z(y) &= \int_H \exp\left(-\frac{1}{2}\|Ax - y\|_\Gamma^2\right)\mu(\mathrm{d}x).
\end{aligned}
\tag{7}
$$

In this context, Baldassari et al. (2024a) defines the conditional infinite-dimensional score as follows:

**Definition 2.3.** Define the (weighted) conditional score function $s(x, t; \mu^y)$ for $x \in H$ as

$$
s(x, t; \mu^y) = \frac{e^t}{1 - e^t}\left(x - e^{\frac{-t}{2}}\mathbb{E}(X_0^\mu | Y = y, X_t^\mu = x)\right).
$$

*Remark* 2.4. Note the dependency of the conditional score on the measurement $y$. In the same spirit of Equation 3, Baldassari et al. (2024a) assume the uniform boundedness in time of the expected square norm of $s(x, t; \mu^y)$ for all $y \in \mathbb{R}^m$ to prove the well-posedness of the corresponding time-reversal SDE, given by Equation 4 with $\mu$ replaced by $\mu^y$ and $s(x, t; \mu)$ by $s(x, t; \mu^y)$.

# 3 UCoS: theory and implementation

In this section, we establish the theoretical foundation for our method by showing that the conditional score associated with the posterior distribution in the linear inverse problem in Equation 6 can be expressed as an affine transformation of an unconditional score derived from a modified diffusion process. This key observation allows us to shift the complexity of posterior sampling to an offline training phase, where only the prior distribution and forward model are used.

## 3.1 UCoS in finite-dimensional space

To build intuition, we begin by illustrating the idea in the finite-dimensional setting $H = \mathbb{R}^n$ and assume $\mathcal{C} = I$ for simplicity. Consider the posterior $\mu^y$ in Equation 7 diffused by Equation 1. Suppose the probability densities of the prior $\mu$ and $X_t^{\mu^y}$ are absolutely continuous w.r.t. Lebesgue measure and denote them by $p_0$ and $q_t(\cdot|y)$, respectively. The transitional probability for the multivariate OU process is given by $\rho_t(x|x_0) = \mathcal{N}(e^{-t/2}x_0, (1 - e^{-t})\mathcal{C})$ and thus

$$
\begin{aligned}
q_t(x|y) &= \int_{\mathbb{R}^n} q_0(z|y)\rho_t(x|z)dz \\
&= \frac{1}{Z(y)}\int_{\mathbb{R}^n} p_0(z)\underbrace{\exp\left(-\frac{1}{2}\|Az - y\|_\Gamma^2\right)\rho_t(x|z)}_{\text{complete square}}dz \\
&= \frac{\omega_t(x, y)}{Z(y)}\int_{\mathbb{R}^n} p_0(z)\exp\left(-\frac{1}{2}\|z - m_t(x, y)\|_{\Sigma_t}^2\right)dz,
\end{aligned}
\tag{8}
$$

where we completed the square w.r.t. the product of likelihood and transitional probability $\rho_t$. Here, the quantities $\Sigma_t$, $m_t(x, y)$ and $\omega(x, y)$ satisfy

$$
\begin{aligned}
\Sigma_t &= \left(\frac{1}{e^t - 1}I + A^\top\Gamma^{-1}A\right)^{-1}, \\
m_t(x, y) &= \Sigma_t\left(A^\top\Gamma^{-1}y + \frac{x}{e^{t/2} - e^{-t/2}}\right) \text{ and} \\
\omega_t(x, y) &\propto \exp\left(-\frac{1}{2}\left\|y - e^{t/2}Ax\right\|_{C_t}^2\right),
\end{aligned}
$$

where $C_t = (e^t - 1)AA^\top + \Gamma \in \mathbb{R}^{m \times m}$. Consequently, we have that

$$
\begin{aligned}
\nabla_x \log q_t(x|y) &= -\frac{1}{2}\nabla_x \left\| y - e^{t/2}Ax \right\|_{C_t}^2 + \nabla_x \log\left((p_0 * \mathcal{N}(0, \Sigma_t))(m_t(x, y))\right) \\
&= -\frac{1}{2}\nabla_x \left\| y - e^{t/2}Ax \right\|_{C_t}^2 + (\nabla_x m_t(x, y))^\top \left[\nabla_x \log\left(p_0 * \mathcal{N}(0, \Sigma_t)\right)\right](m_t(x, y)).
\end{aligned}
\tag{9}
$$

The identity in Equation 8 can be formally interpreted as expressing $q_t$ for a given time as a mixture of the prior $p_0$ and a Gaussian distribution with time-dependent covariance $\Sigma_t$ modulo transformations with $\omega_t$ and $m_t$. What is more, the conditional score $s(\cdot; \mu^y)$ corresponding to $q_t$ in Equation 9 can be expressed as a transformation of the score $\tilde{s}$ of this mixture, $\tilde{s}(x; t; \mu) := \Sigma_t \nabla_x \log\left(p_0 * \mathcal{N}(0, \Sigma_t)\right)$. This idea enables us to approximate the posterior score offline up to affine transformations. More precisely, we have the following identity:

$$
\begin{aligned}
s(x, t; \mu^y) &= \lambda(t)\left(\tilde{s}\left(m_t(x, y), t; \mu\right) + (e^t - 1)A^\top C_t^{-1}(y - e^{t/2}Ax)\right) \\
&= \lambda(t)\left(\tilde{s}\left(m_t(x, y), t; \mu\right) + m_t(x, y) - e^{t/2}x\right),
\end{aligned}
\tag{10}
$$

with

$$
\lambda(t) = (e^{t/2} - e^{-t/2})^{-1}.
$$

*Remark* 3.1. The principled idea of transforming from task-dependent unconditional score to the conditional score outlined above generalizes to other linear diffusion models beyond the specific Ornstein-Uhlenbeck process Equation 1 studied here. This raises the important question of how to design an efficient underlying diffusion model to balance the computational effort further in a desirable way, e.g. by temporal or spectral weighting of the diffusion. This consideration is beyond the scope of this paper.

The result of Equation 10 motivates a new way to approximate the score function in a task-dependent manner. Notice first that the term $m_t(x, y)$ depends on the forward map $A$ in a non-trivial and time-dependent way. Therefore, even if training of $\tilde{s}$ can be performed offline and utilized as an approximation in Equation 10, one would still need to evaluate $A$ during the sample generation. In what follows, we propose an approximation scheme for the conditional score that circumvents this issue. Notice that the dependence of $m_t$ on $A$ is mostly through multiplication with $\Sigma_t$. Hence we may learn the function $r$ given by

$$
r(\cdot, t; \mu) := \tilde{s}(\Sigma_t \cdot, t; \mu) + \Sigma_t.
$$

instead of $\tilde{s}$. By substituting $r$ into Equation 10, we obtain the final identity

$$
s(x, t; \mu^y) = \lambda(t)\left(r(\xi_t(x; y) - e^{t/2}x\right)
$$

with $\xi$ given by

$$
\xi_t(x, y) = A^\top \Gamma^{-1} y + \lambda(t)x.
$$

We can compute and save the term $A^\top \Gamma^{-1} y$ to memory using a single adjoint evaluation. For each time step $t > 0$, we add the time-dependent second term to obtain $\xi_t$ without the need for any further forward evaluations during sampling.

*Remark* 3.2. Assume that the prior $p_0 = \mathcal{N}(0, S_0)$ is Gaussian. Then $r$ is given by

$$
r(\zeta, t; \mu) = \left(\frac{1}{e^t - 1}\mathcal{C}^{-1} + A^\top \Gamma^{-1} A + S_0^{-1}\right)^{-1}\mathcal{C}^{-1}\zeta
$$

and is a bounded linear mapping for any $t > 0$. In particular, $r : \mathbb{R}^n \times \mathbb{R} \to \mathbb{R}^n$ depends only on a spatial variable $\zeta$ and a temporal variable $t$ but **not** on the measurement $y$. This is in stark contrast to the conditional approach Baldassari et al. (2024a), where the conditional score

$$
s : (x, y, t) \mapsto s(x, t; \mu^y) : \mathbb{R}^n \times \mathbb{R}^m \times \mathbb{R} \to \mathbb{R}^n
$$

(see Definition 2.3) depends on **both** spatial variables $x$ and $y$. The added dependence on $y$ increases the input dimensionality from $n + 1$ to $n + m + 1$, which substantially raises the required sample complexity and the size of the neural network required to achieve a given error tolerance.

## 3.2 Theoretical foundations of UCoS in function space

We will now make the transformations above precise in infinite-dimensional separable Hilbert space $H$. First, we define a random process $\{\widetilde{X}_t^\mu\}_{t=0}^T$ such that its distribution is given by the mixture model

$$\widetilde{X}_t^\mu = \widetilde{X}_0^\mu + Z_t, \quad \widetilde{X}_0^\mu \sim \mu \text{ and } Z_t \sim \mathcal{N}(0, \Sigma_t) \tag{11}$$

with $\Sigma_t$ given by

$$\Sigma_t = (e^t - 1)\mathcal{C} - (e^t - 1)^2 \mathcal{C} A^* C_t^{-1} A \mathcal{C} \tag{12}$$

and

$$C_t = (e^t - 1)A\mathcal{C}A^* + \Gamma \in \mathbb{R}^{m \times m}. \tag{13}$$

Notice that $\Sigma_t$ coincides with its formulation in Section 3.1 in finite-dimensional case, see Lemma B.6. The next lemma ensures that $\Sigma_t$ is a valid covariance operator.

**Lemma 3.3.** *For $t > 0$, the operator $\Sigma_t : H \to H$ is trace class, self-adjoint and positive definite.*

*Remark* 3.4. While the mixture in Equation 11 is well-defined for any $t > 0$, we observe that the process $\widetilde{X}_t^\mu$ is no longer diffusive in the sense that it may stay dependent of $\widetilde{X}_0^\mu$ as $t$ grows. Indeed, we notice the dual behavior from two cases:

- if $A = I$, we have $\Sigma_t = \mathcal{C}(\mathcal{C} + (e^t - 1)^{-1}\Gamma)^{-1}\Gamma$.

- for $A = 0$ we have $\Sigma_t = (e^t - 1)\mathcal{C}$.

This indicates different asymptotics for the variance of $\widetilde{X}_t^\mu$ depending on the singular values of $A$. In order to improve intuition on the correlation structure of $\tilde{X}_t$, we depict the diagonal of $\Sigma_t$ for different values of $t$ and the inverse problems related to inpainting in Figure 1 and CT imaging in Figure 2. A detailed description of these problems can be found in Section 5. We further note studying an SDE corresponding to $\widetilde{X}_t^\mu$ or its time-reversal is beyond the scope of this paper.



Figure 1: Diagonal of $\Sigma_t$ for inpainting problem. Time-steps from left to right: $t = 0.05, t = 0.1, t = 0.5$ and $t = 0.9$.

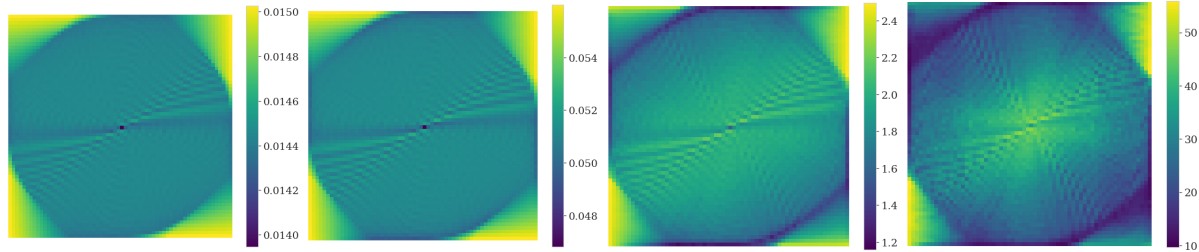

Figure 2: Diagonal of $\Sigma_t$ for CT imaging problem. Time-steps from left to right: $t = 0.05, t = 0.1, t = 0.5$ and $t = 0.9$.

**Definition 3.5.** We define the (weighted, unconditional) score function of the process in Equation 11 by

$$\tilde{s}(z,t;\mu) = -\left(z - \mathbb{E}(\widetilde{X}_0^\mu | \widetilde{X}_t^\mu = z)\right). \tag{14}$$

**Lemma 3.6.** *Assume that $H = \mathbb{R}^n$ and our prior measure $\mu$ admits a density given by $p_0$. Moreover, let $\widetilde{X}_t^\mu$ be defined by Equation 11. Then for every $t \geq 0$, the random variable $\widetilde{X}_t^\mu$ admits a density $\tilde{p}_t(x)$ satisfying*

$$\tilde{s}(z,t;\mu) = \Sigma_t \nabla_z \log \tilde{p}_t(z).$$

Let us fix some $t > 0$ and define the operator $R_t : H \to H$ through

$$R_t := (e^t - 1)I + (e^t - 1)^2 \mathcal{C} A^* C_t^{-1} A. \tag{15}$$

By Lemma B.1, $R_t$ is bijective. Define the affine transformation $r$ of $\tilde{s}$ by

$$r(\cdot, t; \mu) := \tilde{s}(R_t \cdot, t; \mu) + R_t \cdot, \tag{16}$$

Let us now proceed to the main result of this section.

**Theorem 3.7.** *Let $H_\mathcal{C}$ be the Cameron-Martin space of $\mathcal{C}$ (see Appendix A) and assume that the prior satisfies $\mu(H_\mathcal{C}) = 1$. Then the for conditional score function $s(x,t;\mu^y)$ it holds that*

$$s(x,t;\mu^y) = \lambda(t)\left(r\left(\xi_t(x,y)\right) - e^{t/2}x\right) \tag{17}$$

*for $(x,y) \in H \times \mathbb{R}^m$ a.e. in $\mathscr{L}(X_t^\mu, Y)$ and $t > 0$, where*

$$\lambda(t) = (e^{t/2} - e^{-t/2})^{-1}, \qquad \xi_t(x,y) = \mathcal{C} A^* \Gamma^{-1} y + \lambda(t)x. \tag{18}$$

**Special case of Gaussian prior** Let us now study the score $\tilde{s}$ corresponding to the special case of a Gaussian prior measure. This enables us to derive an explicit formula which can give insights, in particular, regarding the regularity of the score function.

**Lemma 3.8.** *Let $\mu = \mathcal{N}(0, S_0)$ be a Gaussian measure and suppose the covariance operator $\widetilde{C}$ satisfies the assumptions of Theorem 3.7. Then there exists a covariance operator $\mathcal{C}$ on $H$ such that $\Sigma_t(\Sigma_t + S_0)^{-1} : H \to H$ can be well-defined as a linear and bounded operator for $t > 0$. Moreover, in that case*

$$\tilde{s}(z,t;\mu) = -\Sigma_t(\Sigma_t + S_0)^{-1}z.$$

## 3.3 Matrix-free implementation

Building on the theoretical framework established in the previous section, we now describe how UCoS can be implemented efficiently in practice. A key strength of our approach is that it remains entirely matrix-free: it requires only the ability to evaluate the forward operator $A$ and its adjoint $A^*$ without ever forming or inverting these operators explicitly. In a discretized setting, this corresponds to avoiding computing and saving the matrix representing $A$, which can be expensive in terms of computational time and memory for large scale problems. This property enables UCoS to scale to high-dimensional and ill-posed inverse problems where standard representations (e.g., via SVD or matrix factorizations) become computationally infeasible. In what follows, we outline the two-phase implementation pipeline: an offline training phase in which a task-dependent score approximation is learned, and an online sampling phase where posterior samples are generated using only the trained network and a precomputed observation-dependent shift.

### 3.3.1 Offline phase: learning the task-dependent score

The next lemma adapts the training procedure discussed in Section 2.1 to the setting of forward process $\widetilde{X}^\mu$. Let us define continuous-time score matching objective (loss function) and the denoising score-matching objective

$$\widetilde{\mathrm{SM}}(r_\theta) := \mathbb{E}\left[\lambda(t)^2 \left\|\tilde{s}(\tilde{x}_t, t; \mu) + \tilde{x}_t - r_\theta(R_t^{-1}\tilde{x}_t, t; \mu)\right\|_H^2\right],$$

$$\widetilde{\text{DSM}}(r_\theta) := \mathbb{E}\left[\lambda(t)^2 \left\| r_\theta(R_t^{-1}\tilde{x}_t, t; \mu) - \tilde{x}_0 \right\|_H^2\right].$$

with the expectations taken w.r.t. $t \sim U[\delta, T], (\tilde{x}_0, \tilde{x}_t) \sim \mathscr{L}(\widetilde{X}_0^\mu, \widetilde{X}_t^\mu)$. We make an assumption regarding the boundedness of the second moment of the conditional score uniformly in time following Baldassari et al. (2024a).

**Assumption 3.9.** The prior $\mu$ has bounded second moment, $\mathbb{E}_{X \sim \mu}\|X\|_H^2 < \infty$, and

$$\sup_{t \in [\delta, T]} \mathbb{E}_{y \sim \pi_y} \mathbb{E}_{x_t \sim \mathscr{L}(X_t^{\mu^y})} \|s(x_t, t; \mu^y)\|_H^2 < \infty.$$

Notice that Assumption 3.9 does not require us to sample from $Y \sim \pi_y$ or to compute the transform $m_t(x, y)$. See Lemma B.4 in the appendix for detailed justification.

**Lemma 3.10.** *Let Assumption 3.9 hold. It follows that*

$$\widetilde{\text{SM}}(r_\theta) = \widetilde{\text{DSM}}(r_\theta) + V,$$

*where the constant $V < \infty$ is independent of $\theta$.*

The result follows from repeating the arguments in Baldassari et al. (2024a). Truncation $t \geq \delta$ guarantees that complications with singularity at $t \to 0$ are avoided, see Appendix B.4 for details.

By the definition of the process $\widetilde{X}_t^\mu$, we have $\widetilde{X}_t^\mu = \widetilde{X}_0^\mu + Z_t$ in distribution, with $Z_t \sim N(0, \Sigma_t)$. Using this identity, we observe that $R_t^{-1}\tilde{x}_t$ conditioned on $\tilde{x}_0$ is Gaussian with covariance

$$\text{Cov}(R_t^{-1}\widetilde{X}_t^\mu \mid \widetilde{X}_0^\mu) = R_t^{-1}\Sigma_t \left(R_t^{-1}\right)^* = \frac{1}{e^t - 1}\mathcal{C} + \mathcal{C}A^*\Gamma^{-1}A\mathcal{C}$$

by the identity

$$R_t^{-1} = \frac{1}{e^t - 1}I + \mathcal{C}A^*\Gamma^{-1}A,$$

followed from Lemma B.1 and Lemma B.6 (i). Now we can directly generate the desired quantity by observing that

$$R_t^{-1}\tilde{x}_t = \left(\frac{1}{e^t - 1}I + \mathcal{C}A^*\Gamma^{-1}A\right)\tilde{x}_0 + \frac{1}{\sqrt{e^t - 1}}\mathcal{C}^{1/2}z_1 + \mathcal{C}A^*\Gamma^{-1/2}z_2$$

in distribution, where $(\tilde{x}_0, \tilde{x}_t) \sim \mathcal{L}(\widetilde{X}_0^\mu, \widetilde{X}_t^\mu)$ and $z_1, z_2 \sim N(0, I)$ i.i.d. In consequence, the training phase requires application of the forward operator $A$, its adjoint $A^*$, the covariance of the noising process $\mathcal{C}$ and the precision matrix of the observational noise $\Gamma^{-1}$. We note that $\Gamma$ in the case of degenerative noise distribution, $\Gamma^{-1}$ can be replaced by the pseudoinverse.

### 3.3.2 Online phase: posterior sampling without forward map evaluations

The offline training, detailed above, will yield an (e.g. FNO) approximation $r_\theta \approx r$ independent of any measurement data. Once this is done, sampling from the posterior is performed by simulating a reverse-time diffusion process using the transformed score $s$, defined as (taking into account Theorem 3.7)

$$s(x, y; \mu^y) = \lambda(t)\left(r_\theta(\xi_t(x, y), t; \mu) - e^{-\frac{t}{2}}x\right),$$

where

$$\xi_t(x, y) = \mathcal{C}A^*\Gamma^{-1}y + \lambda(t)x.$$

Note that after collecting a new measurement vector $y$, the term $\mathcal{C}A^*\Gamma^{-1}y$ is computed once before the generative process and remains constant throughout it. In consequence, the generative process does not require evaluation of any operators in addition to $r_\theta$.

## 4 Convergence analysis

The aim of this section is to conduct rigorously the theoretical convergence of the proposed method. More precisely, we establish a quantitative bound of the error term that indicates how far the samples generated by UCoS lie from the true posterior target measure. In particular, we quantify how this error term depends on different types of numerical approximations. Let us fix $T > 0$ and $\delta \in (0, T)$. Consider a partition $\{\delta = t_1 \leq \cdots \leq t_n = T\}$ of $[\delta, T]$ with mesh size $\Delta t := \min\{t_i - t_{i+1} | 0 \leq i \leq N - 1\}$ and define $\lfloor t \rfloor := \max\{t_i | 1 \leq i \leq n, \ t_i \leq t\}$. For clarity, we reverse time in Equation 4 and let $W_t^{\mu^y} = Y_{T-t}^{\mu^y}$ for $y$ $\pi_y$-a.e. be the ideal solution satisfying

$$dW_t^{\mu^y} = \left(-\frac{1}{2}W_t^{\mu^y} - s(W_t^{\mu^y}, T - t; \mu^y)\right) dt + \mathcal{C}^{1/2} dB_t \tag{19}$$

initiated at $w_0 = (1 - e^{-T})z + e^{-T/2}x_0$ for $z \sim \mathcal{N}(0, \mathcal{C})$ and $x_0 \sim \mu^y$. Observe that Equation 19 is a 'standard' SDE where $W_t^{\mu^y}$ is independent of the future increments of $B_t$. Moreover, let $v_t$ correspond the numerical approximation to $W_t^{\mu^y}$ satisfying

$$dV_t^{\mu^y} = -\frac{1}{2}V_{\lfloor t \rfloor}^{\mu^y} - \lambda(T - \lfloor t \rfloor)\left[r_\theta\left(\xi_{T-\lfloor t \rfloor}(V_{\lfloor t \rfloor}^{\mu^y}, y), T - \lfloor t \rfloor; \mu\right) - e^{(T - \lfloor t \rfloor)/2}V_{\lfloor t \rfloor}^{\mu^y}\right] dt + \mathcal{C}^{1/2} dB_t$$

and initiated at $v_0 = (1 - e^{-T})z$. Here, the two stochastic processes share the same Wiener process $B_t$ and initialization $z$. In what follows, we consider a discrete-time loss given by

$$\varepsilon_{LOSS} = \mathbb{E}_{t \sim U[\delta, T]} \mathbb{E}_{\tilde{x}_{\lfloor t \rfloor} \sim \mathscr{L}(\tilde{X}_{\lfloor t \rfloor}^\mu)} \lambda(\lfloor t \rfloor)^2 \left\| \tilde{s}(\tilde{x}_{\lfloor t \rfloor}, \lfloor t \rfloor; \mu) + \tilde{x}_{\lfloor t \rfloor} - r_\theta(R_{\lfloor t \rfloor}^{-1}\tilde{x}_{\lfloor t \rfloor}, \lfloor t \rfloor; \mu) \right\|_H^2, \tag{20}$$

We are now ready to state the main result of this section.

**Theorem 4.1.** *Let Assumption 3.9 and the assumptions of Theorem 3.7 hold. Assume further that $s(\cdot, t; \mu)$ and $r_\theta(\cdot, t; \mu)$ are Lipschitz continuous with Lipschitz constant $L_s(\cdot) \in L^2([\delta, T])$. Then,*

$$\mathbb{E}_{y \sim \pi_y} \mathbb{E}_{(w_T, v_{T-\delta}) \sim \mathscr{L}(W_T^{\mu^y}, V_{T-\delta}^{\mu^y})} \|w_T - v_{T-\delta}\|_H^2 \leq M\left(\varepsilon_{NUM} + \varepsilon_{LOSS} + \varepsilon_{INIT} + \delta\right)$$

*where $\varepsilon_{LOSS}$ is given in Equation 20, and*

$$\varepsilon_{INIT} \leq e^{-T} \mathbb{E}_{X \sim \mu} \|X\|_H^2, \ \varepsilon_{NUM} \leq \mathcal{O}(\Delta t),$$

*and the constant $M$ depends on the quantities*

$$\mathbb{E}_{t \sim U[0, T-\delta]} \mathbb{E}_{y \sim \pi_y} \mathbb{E}_{w_t \sim \mathscr{L}(w_t)} \|s(w_t, T - t; \mu^y)\|_H^2, \mathbb{E}_{X \sim \mu} \|X\|_H^2, \ \int_\delta^T L_s^2(\tau) d\tau, \ \mathrm{tr}_H(\mathcal{C}) \ \ and \ \ T.$$

Note that the Lipschitz continuity assumption of the drift is a natural setup for the backward SDE to retain the uniqueness of a (strong) solution, see Pidstrigach et al. (2024). It also follows from there that for $\pi_y$-almost every $y \in \mathbb{R}^m$ the random variable $W_T^{\mu^y}$ is distributed according to the posterior $\mu^y$.

## 5 Numerical experiments

We showcase our method in the context of inverse problems related to a toy inpainting example, computerized tomography (CT) and a deblurring problem. Details of the numerical implementation are described in Appendix D and enlarged depictions of posterior samples can be found in Appendix E.

To enable a principled comparison, we examine the accuracy of posterior sampling using score approximations trained with architectures that offer comparable runtimes. Methods that avoid forward evaluations during sampling, such as UCoS and conditional method, exhibit similar runtimes. However, we find that the performance of the conditional method is hindered by a more complex score approximation, especially when neural operators have very few parameters. In contrast, the computational cost of posterior sampling

with unconditional methods increases due to the need for repeated forward evaluations during the sampling process. When comparing unconditional methods to UCoS, deviations from the true posterior are primarily attributed to the approximative nature of the conditioning on measurement information, rather than inaccuracies in the score approximation. This is because the dimensionality of the score approximation is the same for both UCoS and unconditional methods.

Following Baldassari et al. (2024a), we utilize a Fourier neural operator (FNO) Li et al. (2021) to parameterise $r_\theta$ and perform training outlined in Section 2.1 on an Nvidia A100 GPU with 80 GiB. We compare UCoS to four different methods. First, we consider one of the first conditional score approximation considered in Jalal et al. (2021); Feng et al. (2023) and refer to it by SDE ALD. Additionally, we implement Diffusion Posterior Sampling (DPS) Chung et al. (2023a), projection based methods for general inverse problems (Proj) Dey et al. (2024) and the conditional score approach (Conditional) Baldassari et al. (2024a). Each score approximation is given explicitly in Appendix D. The numerical implementation can be found at `https://github.com/FabianSBD/SBD-task-dependent/tree/UCoS`.

**Inpainting** In this paragraph, the prior is modeled by a Gaussian distribution with squared exponential covariance structure. This allows us to compute all score functions in closed form without any learning. Uniquely to this example, we are able to sample from the (Gaussian) posterior. The forward mapping $A : \mathbb{R}^n \to \mathbb{R}^n$ is diagonal and with diagonal entries one in the center of the image and zero elsewhere. The information on the center pixels is lost and in the inverse problem we seek to recover the full image. The image resolution is $64 \times 64$ pixels.

Figure 3 and Table 1 illustrate that the posterior ensemble ($N = 1000$) generated by UCoS (our method) and conditional method generate samples that are consistent with the true posterior distribution. Since no score needs to be learned, UCoS and Conditional perform very similarly.

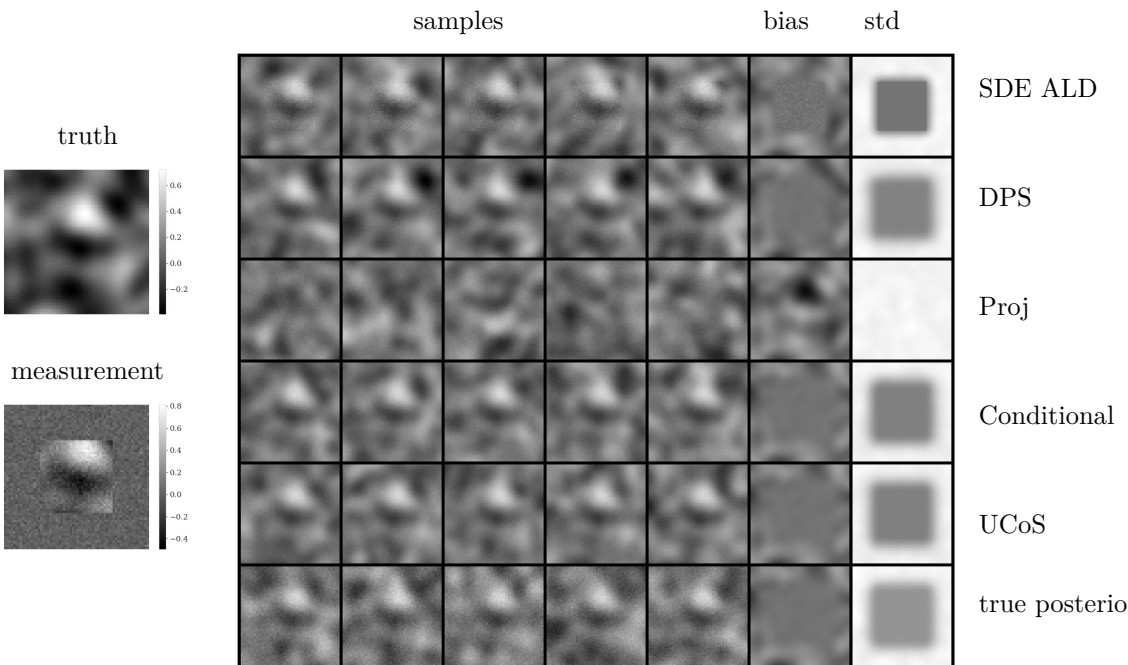

Figure 3: Summary of posterior samples for inpainting problem. On the left column, the top image is ground truth and the bottom is measurement data. On the right column, 5 posterior samples with true score corresponding to a Gaussian prior and their bias and standard deviation, with methods used from top to bottom: SDE ALD, DPS, Proj, Conditional, UCoS and true posterior.

| method | bias | std | time |
|---|---|---|---|
| SDE ALD | 0.1341 | 0.1419 | 202 |
| DPS | 0.1050 | 0.1242 | 467 |
| Proj | 0.2008 | 0.1772 | 254 |
| Conditional | 0.1011 | 0.1238 | 198 |
| UCoS | 0.1003 | 0.1226 | 197 |
| True | 0.1013 | 0.1304 | <1 |

Table 1: Summary statistics of all methods for the inpainting problem. Bias and std are computed on 1000 generated posterior samples and averaged over the number of pixels. Computational time is given in seconds.

**CT imaging**  Here, the forward mapping $A$ models a sparse-view imaging setting with a 45-degree angle of view with 256 equiangular directions. The detector is assumed to have 256 apertures spanning over the width of imaging area (i.e., 256 parallel line integrals per angle are measured) and, consequently, the problem dimensions are given by $m = n = 256^2$. Moreover, the measurement data is corrupted by additive Gaussian noise with signal–to–noise ratio of 20 dB.

Our training data is the Lung Image Database Consortium image collection (LIDC–IRI)–dataset Armato 3rd et al. (2011), containing $> 200.000$ 2D slices of resolution $512 \times 512$, we rescale them to $256 \times 256$.

We analyze the influence of the complexity of the FNO approximation on the posterior samples. We employ an architecture with 4 layers while varying the number of nodes uniformly across all layers. For the full $256 \times 256$ resolution, we analyze posterior samples for an architecture with 32, 64 and 128 neurons per layer in Figure 4 and Table 2. We see that UCoS and conditional method have a significant edge in terms of computational time over the other methods as a consequence of avoiding repeated forward evaluations during sampling. With larger number of neurons per layer, conditional method produces relatively similar samples as UCoS in terms bias and standard deviation, but has worse statistics and visual sample quality when using only 32 neurons. Due to the complexity of the score approximation, the performance of conditional method at 32 nodes per layer differs the most from the performance using more nodes. Sample quality of UCoS seems comparable or better to the unconditional methods (SDE ALD, DPS and Proj) in terms of standard deviation and bias and a naive visual inspection. Noticeably, UCoS generates samples with low variance compared to the other methods.

| method | **bias** for k nodes | | | **std** for k nodes | | | **time** for k nodes | | |
|---|---|---|---|---|---|---|---|---|---|
| | k=32 | k=64 | k=128 | k=32 | k=64 | k=128 | k=32 | k=64 | k=128 |
| SDE ALD | 0.1014 | 0.1042 | 0.1044 | 0.0741 | 0.1064 | 0.0644 | 35 | 48 | 77 |
| DPS | 0.0605 | 0.0552 | 0.0788 | 0.068 | 0.0768 | 0.1723 | 69 | 99 | 176 |
| Proj | 0.0644 | 0.0610 | 0.0616 | 0.0667 | 0.078 | 0.0702 | 224 | 237 | 266 |
| Conditional | 0.2336 | 0.0621 | **0.0467** | 0.0923 | 0.0594 | 0.0516 | **15** | **27** | **57** |
| UCoS | **0.0489** | **0.0498** | 0.0541 | **0.0379** | **0.0316** | **0.0262** | **15** | **27** | **57** |

Table 2: Summary statistics of all methods for the CT imaging problem. Left table: bias, middle: std and right table: computational time in minutes. Bias and std are computed on 1000 generated posterior samples and averaged over the number of pixels. In each table, the columns correspond to the number of neurons used per layer.

In addition to the previous experiment, for UCoS and conditional method, we train models for numerous number of neurons on a reduced $64 \times 64$ resolution. In Figure 5, we compare the number of nodes to the resulting posterior bias and standard deviation for the full $256 \times 256$ resolution (crosses) and the reduced $64 \times 64$ resolution (lines). We observe that, especially in terms of standard deviation, UCoS achieves saturation with lower model complexity. To provide context, halving number of nodes approximately halves the online sampling time.

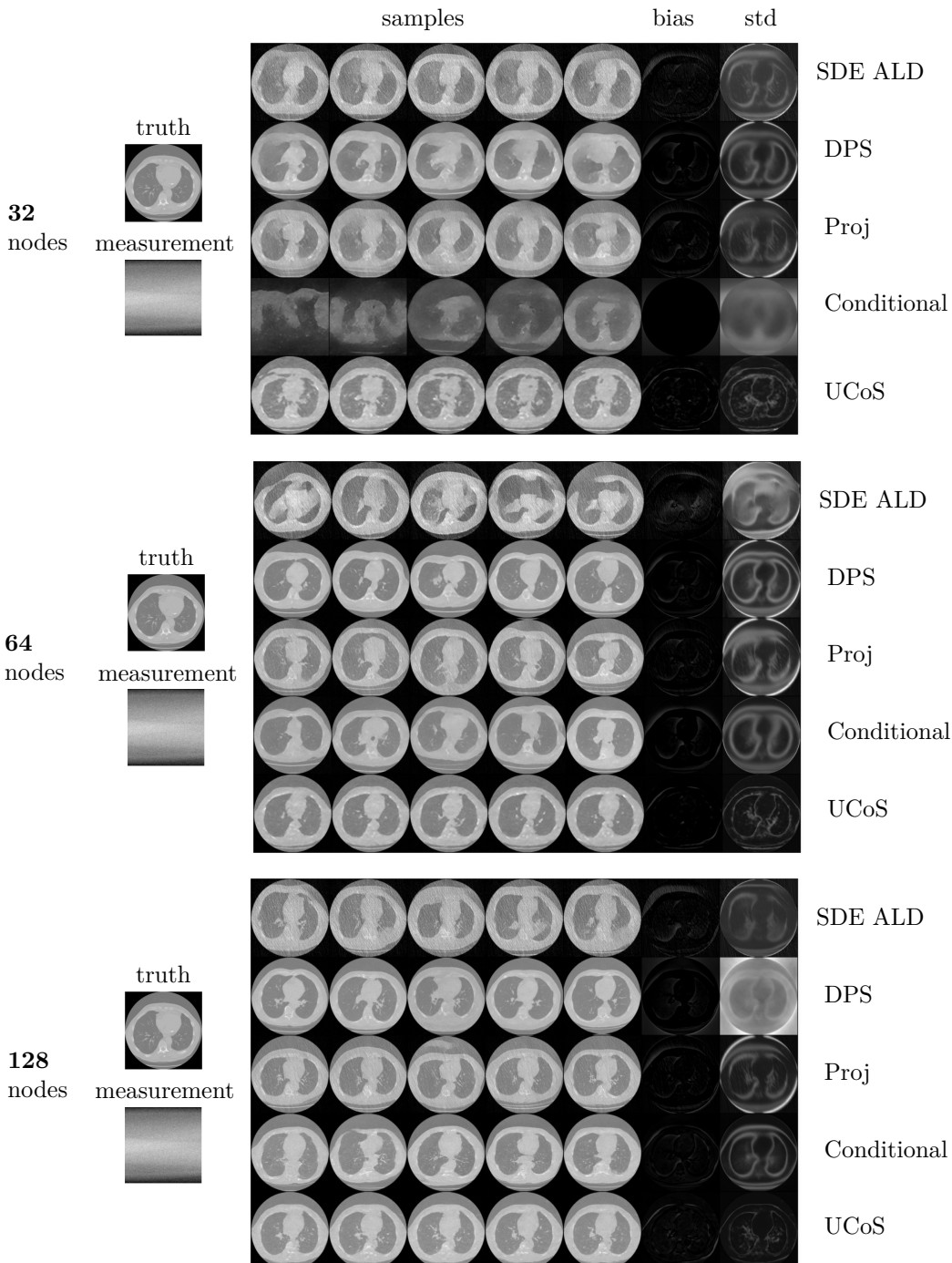

Figure 4: Summary of posterior samples for CT imaging problem. Top figure uses FNO with 32 nodes per layer, middle figure uses 64 nodes per layer and bottom row 128 nodes per layer. In each figure, on the left column, the top image is ground truth and the bottom is measurement data (sinogram). Both are fixed for all FNO architectures. On the right column, 5 posterior samples, their bias and standard deviation. Methods used from top to bottom: SDE ALD, DPS, Proj, Conditional, and UCoS.

**Deblurring** In this example, the forward operator A corresponds to a two-dimensional convolution with a Gaussian kernel over the image domain. We observe the output function on the same grid with an additive Gaussian noise vector.

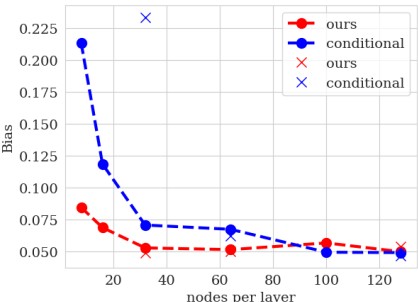 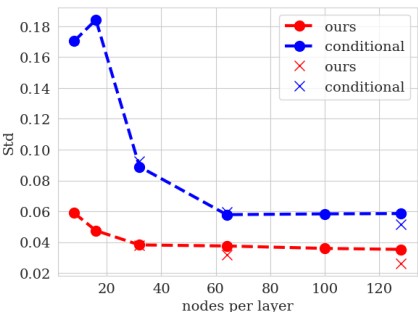

Figure 5: Comparison of the error dependence of the conditional method and UCoS on the parametrization of the score approximation. X-axis: number of nodes per layer in the FNO, Y-axis: L2 norm of the bias (left) or std (right), averaged over the number of pixels. Dashed lines correspond to $64 \times 64$ resolution and crosses correspond to the problem with $256 \times 256$ resolution.

The training data is given by the Large-scale CelebFaces Attributes (CelebA) Dataset Liu et al. (2015), which contains $> 200.000$ face images of celebrities. We scale the images to square resolution $175 \times 175$.

| method | bias | std | time |
|---|---|---|---|
| SDE ALD | 0.0878 | 0.0915 | 300 |
| DPS | 0.0872 | 0.0801 | 239 |
| Proj | **0.0785** | 0.0622 | 15822 |
| Conditional | 0.2178 | 0.0950 | **85** |
| UCoS | 0.0884 | **0.0345** | **85** |

Table 3: Summary statistics of all methods for the Deblurring problem. Bias and std are computed on 1000 generated posterior samples and averaged over the number of pixels. Computational time is given in minutes.

We demonstrate in Figure 6 and Table 3 that UCoS and conditional achieve the fastest sampling speed. For the chosen FNO parameterization, the conditional method produces samples with higher bias and standard deviation as well as low visual quality. Visually, the posterior ensemble ($N = 1000$) produced by UCoS is consistent with the ground truth, while other methods introduce artifacts that are inconsistent with human facial features. In terms of bias, UCoS is at a roughly similar or slightly better level than unconditional methods, while the pixel-wise standard deviation is noticeably lower than other methods.

# 6 Conclusion, limitations and future work

This paper addresses the challenge of balancing the computational cost between offline training and online posterior sampling in large-scale inverse problems. In many such problems, evaluating the forward operator during sampling constitutes a major computational bottleneck. We introduce a novel, theoretically grounded method UCoS that entirely removes this burden during sampling by shifting it to the offline training phase, without introducing any approximation error. Our approach combines the computational benefits of the conditional method introduced in Baldassari et al. (2024a) with a more scalable score-learning formulation. Furthermore, we demonstrate that this method is rigorous in an infinite-dimensional setting and, therefore, independent of discretization.

We validate the correctness of our approach with numerical experiments related to CT imaging and Deblurring. UCoS generates samples with approximately comparable or better quality than unconditional approaches while offering tangible computational benefits during sampling. However, it remains task-dependent, requiring re-training when the forward problem is modified. The quality of posterior samples is assessed using simple statistics such as bias, pixel-wise standard deviation, and visual inspection. Hence, the

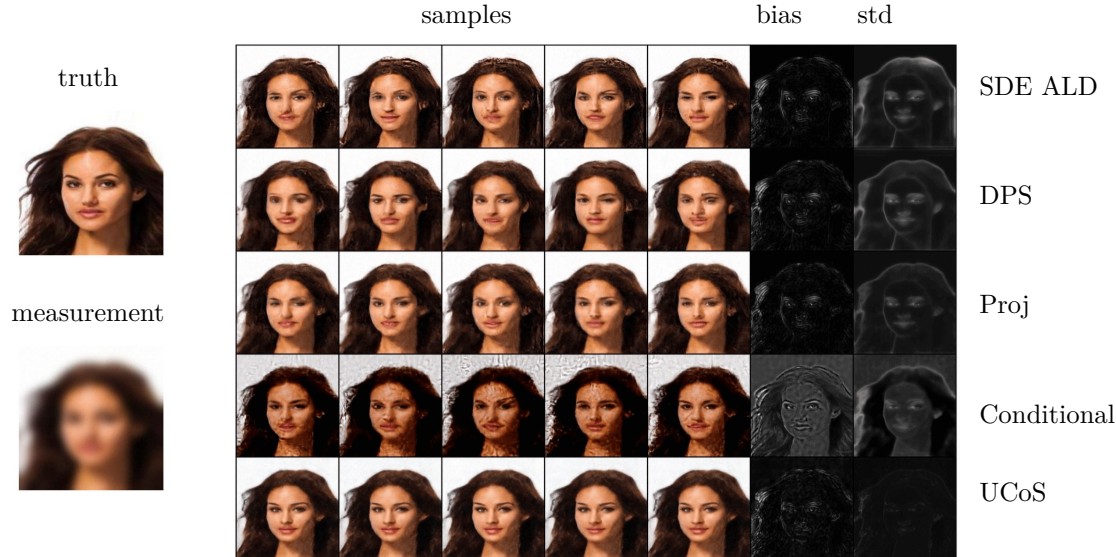

Figure 6: Summary of posterior samples for the Deblurring problem. On the left column, the top image is ground truth and the bottom is measurement data. On the right column, 5 posterior samples and their bias and standard deviation, with methods used from top to bottom: SDE ALD, DPS, Proj, Conditional, and UCoS.

results should be viewed as proof of concept. Further experiments are needed for a more comprehensive comparison of sample quality. While UCoS does not accelerate posterior sampling compared to the conditional method when using score approximation with similar complexity, we show that conditional methods, unlike UCoS, involve more complex function approximations. This complexity can hinder performance, particularly for neural operators with a limited number of parameters.

The general principle of transforming a task-dependent unconditional score into a conditional score, as developed in this work, extends beyond the specific OU diffusion process considered here. This opens up intriguing questions about selecting the appropriate process to achieve a more efficient balance of computational effort.

That said, the theoretical foundation of UCoS relies critically on the interplay between a linear forward model and a Gaussian likelihood, which together enable the diffused posterior to be reinterpreted as a diffused prior under a new, task-specific process. This structure allows us to derive an exact identity between the conditional score and a transformed unconditional score, as captured in Theorem 3.7. However, extending this exact correspondence beyond settings where one can 'complete the square' – as done in Equation 8 – appears nontrivial. Investigating whether similar structures can be identified in nonlinear or non-Gaussian problems remains an important avenue for future work.

## Acknowledgements

This work has been supported by the Research Council of Finland (RCoF) through the *Flagship of advanced mathematics for sensing, imaging and modelling* (FAME), decision number 358944. Moreover, TH was supported through RCoF decision numbers 353094 and 348504. M.L. was partially supported by PDE-Inverse project of the European Research Council of the European Union, the FAME and Finnish Quantum flagships and the grant 336786 of the RCoF. Views and opinions expressed are those of the authors only and do not necessarily reflect those of the European Union or the other funding organizations. Neither the European Union nor the other funding organizations can be held responsible for them. MVdH gratefully acknowledges the support of the Department of Energy BES, under grant DE-SC0020345, Oxy, the corporate members of the Geo-Mathematical Imaging Group at Rice University and the Simons Foundation under the MATH + X Program. The authors acknowledge the National Cancer Institute and the Foundation for the

National Institutes of Health, and their critical role in the creation of the free publicly available LIDC/IDRI Database used in this study.

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

# A  Probability measures on Hilbert spaces

## A.1  Gaussian random processes in an infinite-dimensional Hilbert space

This section introduces notations and outlines some basic properties of probability measures on Hilbert spaces. For a more comprehensive introduction, we refer to Da Prato & Zabczyk (2014); Hairer (2023).

**Gaussian measures on Hilbert space**  Let $(H, \langle \cdot, \cdot \rangle_H)$ be a separable Hilbert space with norm $\| \cdot \|_H = \sqrt{\langle \cdot, \cdot \rangle_H}$. A bounded linear operator $C : H \to H$ is called *self-adjoint* if $\langle x, Cy \rangle_H = \langle Cx, y \rangle_H$ for all $x, y \in H$ and *positive definite* if $\langle Cx, x \rangle_H \geq 0$ for all $x \in H$. We say that a self-adjoint and positive definite operator $C$ is of *trace class* if

$$\mathrm{tr}_H(C) := \sum_{n=1}^{\infty} \langle Ce_n, e_n \rangle < \infty,$$

where $\{e_n\}$ is an orthogonal basic of $H$. We denote by $L_1^+(H)$ the space of all self-adjoint, positive definite and trace class operators on $H$. A random variable taking values in $H$ is called *Gaussian* if the law of $\langle h, X \rangle_H$ is Gaussian for each $h \in H$. Gaussian random variables are determined by their *mean* $m = \mathbb{E}[X] \in H$ and their *covariance operator* defined as

$$\langle Cg, h \rangle = \mathbb{E} \left[ \langle g, X - m \rangle \langle h, X - m \rangle \right].$$

In this case, we denote $X \sim \mathcal{N}(m, C)$. If $m = 0$, $X$ is called *centred*. It can be shown that if $X$ is a Gaussian random variable on $H$ then $C \in L_1^+(H)$, moreover, $\mathbb{E}[\|X\|_H^2] = \mathrm{tr}_H(C)$.

**The Cameron–Martin space**  We define the Cameron–Martin space associated with a Gaussian measure $\mu = \mathcal{N}(0, C)$ on $H$ to be the intersection of all linear spaces of full measure under $\mu$, and denote it by $H_\mu$ or $H_C$. It can be shown that $H_\mu = C^{1/2} H$ and $H_\mu$ is compactly embedded and dense in $H$. In infinite dimensions, it is necessarily the case that $\mu(H_\mu) = 0$. Moreover, $H_\mu$ can be endowed with a Hilbert space structure with an inner product

$$\langle g, h \rangle_{H_\mu} = \langle g, C^{-1} h \rangle_H = \langle C^{-1/2} g, C^{-1/2} h \rangle_H.$$

*Example* 1. Let $H = \mathbb{R}^d$ and $\mu = \mathcal{N}(0, C)$ be a Gaussian measure on $H$ with a positive definite covariance matrix $C \in \mathbb{R}^{d \times d}$. Then since $C^{1/2} H = H$, the Cameron–Martin space is the whole space $\mathbb{R}^d$.

**Cameron–Martin's theorem**  The Cameron–Martin space $H_\mu$ plays a special role in that it characterises precisely those directions in which one can translate the Gaussian measure $\mu$ without changing its null sets, thus $\mu_h := \mathcal{N}(h, C)$ and $\mu = \mathcal{N}(0, C)$ are equivalent if and only if $h \in H_\mu$. Moreover, the Radon–Nikodym derivative of $\mu_h$ with respect to $\mu$ is given by

$$\frac{\mathrm{d}\mu_h}{\mathrm{d}\mu}(x) = \exp\left( \langle h, x \rangle_{H_\mu} - \frac{1}{2} \|h\|_{H_\mu}^2 \right), \quad \mu\text{-a.s.}, x \in H. \tag{21}$$

Note that since $H_\mu = C^{1/2} H$ is dense in $H$, the random variable $\langle h, x \rangle_{H_\mu} = \langle C^{-1/2} h, C^{-1/2} x \rangle_H$, $x \in H$, can be defined properly using a limiting process, see Remark 2.24 in Da Prato & Zabczyk (2014).

## A.2  Wiener processes $B_t$ in an infinite-dimensional Hilbert space

In this subsection, we recall the definition of the Wiener processes $B_t$ in an infinite-dimensional Hilbert space. As $H$ is an infinite-dimensional Hilbert space, the process $B_t$ can not be defined as random variable taking values in the Hilbert space $H$, but we need to start our considerations with a larger Hilbert space $Y$ having the additional properties that are described below. At each time $t \in [0, T]$ the process $B_t$ is defined to be a Gaussian random variable taking values in the Hilbert space $Y$ so that the expectation of $B_t$ is zero and the covariance operator is $tC_Y$, where $C_Y : Y \to Y$ is a symmetric and injective trace-class operator in

$Y$. That is, for $\phi, \psi \in Y$ and $0 \leq t < t + s \leq T$ we have

$$
\begin{aligned}
&\mathbb{E}(\langle B_t, \phi \rangle_Y) = 0, \\
&\mathbb{E}(\langle B_t, \phi \rangle_Y \cdot \langle B_t, \psi \rangle_Y) = t \langle \phi, C_Y \phi \rangle_Y, \\
&\mathbb{E}(\langle B_{t+s} - B_t, \phi \rangle_Y \cdot \langle B_{t+s} - B_t, \psi \rangle_Y) = s \langle \phi, C_Y \phi \rangle_Y.
\end{aligned}
$$

We require that $Y$ and $C_Y$ are such that

$$
H = \mathrm{Ran}(C_Y^{1/2}) = C_Y^{1/2}(Y).
$$

Under the above assumptions, $H$ is the Cameron-Martin of the Gaussian random variable $B_t$ for $t > 0$. Then, we can define for $b \in Y$ and $\phi \in C_Y(Y)$ an extension of $H$-inner product $\langle u, \phi \rangle_H$ by setting

$$
\langle u, \phi \rangle_H := \langle u, C_Y^{-1} \phi \rangle_Y.
$$

Using these definitions, it holds that for all $\phi, \psi \in C_Y(Y) \subset H$ and $0 \leq t < t + s \leq T$ we have

$$
\begin{aligned}
&\mathbb{E}(\langle B_t, \phi \rangle_H) = 0, \\
&\mathbb{E}(\langle B_t, \phi \rangle_H \cdot \langle B_t, \psi \rangle_H) = t \langle \phi, \phi \rangle_H, \\
&\mathbb{E}(\langle B_{t+s} - B_t, \phi \rangle_H \cdot \langle B_{t+s} - B_t, \psi \rangle_H) = s \langle \phi, \phi \rangle_Y.
\end{aligned}
$$

Motivated by these formulas, we call $B_t$, $0 \leq t \leq T$, a Wiener process in $H$ having the (generalized) covariance operator $tI$.

We also note that in formula Equation 1, we use the symmetric trace-class operator $\mathcal{C} \colon H \to H$ and the increments $\mathcal{C}^{1/2} dB_t$. These increments can be interpreted as the differences of the random process $t \mapsto \mathcal{C}^{1/2} B_t$. At any time $0 \leq t \leq T$, the random variable $\mathcal{C}^{1/2} B_t$ takes values in the space $H$, and its covariance operator in $H$ is $\mathcal{C} \colon H \to H$.

## B  Proofs for Section 3

### B.1  Task-depedent score

Recall that $C_t = \Gamma + (e^t - 1) A \mathcal{C} A^* \colon H \to H$ and $\Sigma_t = (e^t - 1)\mathcal{C} - (e^t - 1)^2 \mathcal{C} A^* C_t^{-1} A \mathcal{C} \colon H \to H$.

*Proof of Lemma 3.3.* It is clear that $\Sigma_t$ is self-adjoint. For positive definiteness, we write

$$
\begin{aligned}
\Sigma_t &= \mathcal{C}^{1/2} \left( (e^t - 1)I - (e^t - 1)^2 \mathcal{C}^{1/2} A^* C_t^{-1} A \mathcal{C}^{1/2} \right) \mathcal{C}^{1/2} \\
&= \mathcal{C}^{1/2} \left( \frac{1}{e^t - 1} I + \mathcal{C}^{1/2} A^* \Gamma^{-1} A \mathcal{C}^{1/2} \right)^{-1} \mathcal{C}^{1/2},
\end{aligned}
$$

where we applied Lemma B.1 (a) for the last identity. Since $\mathcal{C} A^* C_t^{-1} A \mathcal{C}$ is also positive definite, it immediately follows that

$$
0 < \mathrm{tr}_H(\Sigma_t) = (e^t - 1)\mathrm{tr}_H(\mathcal{C}) - (e^t - 1)^2 \mathrm{tr}_H(\mathcal{C} A^* C_t^{-1} A \mathcal{C} \leq (e^t - 1)\mathrm{tr}_H(\mathcal{C}) < \infty
$$

and, consequently, $\Sigma_t$ is trace-class. $\qquad\square$

*Proof of Lemma 3.6.* We replicate the proof of Lemma 1 in Pidstrigach et al. (2024) for the process $\widetilde{X}_t^\mu$. The density $\tilde{p}_t(\cdot | x_0)$ of $\widetilde{X}_t^\mu$ conditioned on $\widetilde{X}_0^\mu = x_0$ is Gaussian centered at $x_0$ with covariance $\Sigma_t$.

For what follows, let $\tilde{p}_0(x_0)$ be the density of $\widetilde{X}_0^\mu$ and $\tilde{p}_t(x)$ the density of $\widetilde{X}_t^\mu$. We apply Leibniz's rule to obtain

$$
\begin{aligned}
\Sigma_t \nabla_x \log \tilde{p}_t(x) &= \Sigma_t \left( \frac{1}{\tilde{p}_t(x)} \nabla_x \int \tilde{p}_t(x|x_0) d\tilde{p}_0(x_0) \right) \\
&= -\Sigma_t \int \Sigma_t^{-1}(x - x_0) \frac{\tilde{p}_t(x|x_0)}{\tilde{p}_t(x)} d\tilde{p}_0(x_0) \\
&= -\int (x - x_0) \tilde{p}(x_0|x) \\
&= -\left( x - \mathbb{E}(\widetilde{X}_0^\mu | \widetilde{X}_t^\mu = x) \right),
\end{aligned}
$$

where we utilized Bayes' formula for the second last identity. This proves the claim. $\qquad\square$

**Lemma B.1.** *For any $t > 0$, we have that*

*(a) the linear operator $\Xi_t = (e^t - 1)I - (e^t - 1)^2 \mathcal{C}^{1/2} A^* C_t^{-1} A \mathcal{C}^{1/2} : H \to H$ is bijective and*

$$
\Xi_t^{-1} = \frac{1}{e^t - 1} I + \mathcal{C}^{1/2} A^* \Gamma^{-1} A \mathcal{C}^{1/2},
$$

*(b) the linear operator $\Xi'_t = I - (e^t - 1)\mathcal{C} A^* C_t^{-1} A$ is bijective and*

$$
(\Xi'_t)^{-1} = I + (e^t - 1)\mathcal{C} A^* \Gamma^{-1} A; \quad and
$$

*(c) it holds that*

$$
(e^t - 1)\mathcal{C} A^* C_t^{-1} = \Sigma_t A^* \Gamma^{-1}.
$$

*Proof.* a) Invertibility from the right can be derived by straightforward computation

$$
\Xi_t \left( \frac{1}{e^t - 1} I + \mathcal{C}^{1/2} A^* \Gamma^{-1} A \mathcal{C}^{1/2} \right)
$$
$$
= I + (e^t - 1)\mathcal{C}^{1/2} A^* \Gamma^{-1} A \mathcal{C}^{1/2} - (e^t - 1)\mathcal{C}^{1/2} A^* C_t^{-1} \underbrace{\left( (e^t - 1)A \mathcal{C} A^* + \Gamma \right)}_{=C_t} \Gamma^{-1} A \mathcal{C}^{1/2} = I
$$

Similarly, invertibility from the left follows from an analogous computation. The invertibility in (b) can be established using the same arguments. For the identity in (c), we have

$$
\begin{aligned}
\Sigma_t A^* \Gamma^{-1} C_t &= (e^t - 1)\mathcal{C} A^* - (e^t - 1)^2 \mathcal{C} A^* C_t^{-1} A \mathcal{C} A^* + (e^t - 1)^2 \mathcal{C} A^* \Gamma^{-1} A \mathcal{C} A^* \\
&\quad -(e^t - 1)^3 \mathcal{C} A^* C_t^{-1} A \mathcal{C} A^* \Gamma^{-1} A \mathcal{C} A^* \\
&= (e^t - 1)\mathcal{C} A^* + (e^t - 1)^2 \mathcal{C} A^* \Gamma^{-1} A \mathcal{C} A^* \\
&\quad -(e^t - 1)^2 \mathcal{C} A^* C_t^{-1} \underbrace{\left( (e^t - 1)A \mathcal{C} A^* + \Gamma \right)}_{=C_t} \Gamma^{-1} A \mathcal{C} A^* \\
&= (e^t - 1)\mathcal{C} A^*.
\end{aligned}
$$

The desired identity follows by inverting $C_t$. $\qquad\square$

## B.2  Proof of Theorem 3.7

We follow the structure in Section 3.1 and first prove the following equivalent of Equation 10. Theorem 3.7 follows directly from substituting Equation 16 into Theorem B.2 while utilizing Lemma B.8.

**Theorem B.2.** *Let $H_\mathcal{C}$ be the Cameron-Martin space of $\mathcal{C}$ and assume that the prior satisfies $\mu(H_\mathcal{C}) = 1$. For $t > 0$, let $\Sigma_t$ be given by Equation 12 and define*

$$m_t(x, y) = e^{t/2}x + (e^t - 1)\mathcal{C}A^* C_t^{-1}(y - e^{t/2}Ax). \tag{22}$$

*Then the conditional score function $s(x, t; \mu^y)$ related to it holds that*

$$s(x, t; \mu^y) = \lambda(t) \left( \tilde{s}\left(m_t(x, y), t; \mu\right) + m_t(x, y) - e^{t/2}x \right) \tag{23}$$

*for $(x, y) \in H \times \mathbb{R}^m$ a.e. in $\mathscr{L}(X_t^\mu, Y)$ and $t > 0$.*

Theorem B.2 is a direct consequence of the following proposition.

**Proposition B.3.** *Let the assumptions of Theorem 3.7 hold. Then*

$$\mathbb{E}(\widetilde{X}_0^\mu | \widetilde{X}_t^\mu = m_t(x, y)) = \mathbb{E}(X_0^\mu | Y = y, X_t^\mu = x), \tag{24}$$

*for $(x, y) \in H \times \mathbb{R}^m$ a.e. in $\mathscr{L}(X_t^\mu, Y)$ and $t > 0$.*

Throughout this section, we denote the transition kernel densities

$$n_t(x_0, x) := \frac{d\mathcal{N}(x_0, (e^t - 1)\mathcal{C})}{d\mathcal{N}(0, (e^t - 1)\mathcal{C})}(e^{t/2}x), \quad \tilde{n}_t(x_0, x) := \frac{d\mathcal{N}(x_0, \Sigma_t)}{d\mathcal{N}(0, \Sigma_t)}(x),$$

whenever the Radon–Nykodym derivatives make sense.

A plan of proof for Proposition B.3 is as follows. We first develop auxiliary results: Lemma B.4 shows that the laws of $m_t(X_t^\mu, Y)$ and $\widetilde{X}_t^\mu$ coincide when conditioned on $X_0 = 0$. Lemma B.5 is used to express each expectation in Equation 24 in terms of transition kernels $n_t$ and $\tilde{n}_t$ of the corresponding forward SDEs, which in turn can be written as Radon–Nykodym derivatives of certain measures. After that, we use Lemma B.6 to show that the measures $\mathcal{N}(0, (e^t - 1)\mathcal{C})$ and $\mathcal{N}(0, \Sigma_t)$ are equivalent, in particular their Cameron–Martin spaces equal, which concludes the proof. Finally, we put together the argument at the end of the section.

**Lemma B.4.** *Let $Z_1 \sim \mathcal{N}(0, (1 - e^{-t})\mathcal{C})$, $Z_2 \sim \mathcal{N}(0, \Gamma)$ and $Z_3 = \mathcal{N}(0, \Sigma_t)$ be mutually independent Gaussian random variables on $H$. Moreover, let $x_0 \in H$ be arbitrary. It holds that*

$$\mathscr{L}(m_t(e^{-t/2}x_0 + Z_1, Ax_0 + Z_2)) = \mathscr{L}(x_0 + Z_3).$$

*for any $t > 0$.*

*Proof.* We have that

$$
\begin{aligned}
m_t(e^{-t/2}x_0 + Z_1, Ax_0 + Z_2) &= x_0 + e^{t/2}Z_1 + (e^t - 1)\mathcal{C}A^* C_t^{-1}(Z_1 - e^{t/2}AZ_1) \\
&= x_0 + e^{t/2}\frac{1}{e^t - 1}\Sigma_t \mathcal{C}^{-1} Z_1 + \Sigma_t A^* \Gamma^{-1} Z_2
\end{aligned}
$$

is a Gaussian random variable centered at $x_0$ with a covariance

$$
\begin{aligned}
\mathrm{Cov}(m_t(e^{-t/2}x_0 + Z_1, Ax_0 + Z_2)) &= \Sigma_t A^* \Gamma^{-1} A\Sigma_t + \frac{1}{e^t - 1}\Sigma_t \mathcal{C}^{-1}\Sigma_t \\
&= \Sigma_t \left( A^* \Gamma^{-1} A + \frac{1}{e^t - 1}\mathcal{C}^{-1} \right) \Sigma_t = \Sigma_t.
\end{aligned}
$$

This proves the claim. $\qquad\square$

**Lemma B.5.** *The following holds*

(i) *For $(x, y) \in H \times \mathbb{R}^m$ a.e. in $\mathscr{L}(X_t^\mu, Y)$,*

$$\mathbb{E}(X_0^\mu | Y = y, X_t^\mu = x) = \frac{\int_H x_0 n_t(x_0, x)\mu^y(dx_0)}{\int_H n_t(x_0, x)\mu^y(dx_0)}.$$

*(ii) For $x \in H$ $\mathscr{L}(\widetilde{X}_t^\mu)$-a.e.,*

$$\mathbb{E}(\widetilde{X}_0^\mu | \widetilde{X}_t^\mu = x) = \frac{\int_H x_0 \tilde{n}_t(x_0, x) \mu(dx_0)}{\int_H \tilde{n}_t(x_0, x) \mu(dx_0)}.$$

*Proof.* **(i):** We first observe that, if $X \sim \mathcal{N}(x_0, \mathcal{C})$ then $\alpha X \sim \mathcal{N}(\alpha x_0, \alpha^2 S)$ for any $\alpha > 0$, and as a direct consequence

$$\mathcal{N}(x_0, \mathcal{C})(A) = \mathcal{N}(\alpha x_0, \alpha^2 \mathcal{C})(\alpha A), \tag{25}$$

for any $\alpha > 0$ and $A \in \mathcal{B}(H)$. By using this property for $\alpha = e^{-t/2}$, it holds that

$$
\begin{aligned}
n_t(x_0, x) &= \frac{d\mathcal{N}(x_0, (e^t - 1)\mathcal{C})}{d\mathcal{N}(0, (e^t - 1)\mathcal{C})}(e^{t/2} x) \\
&= \frac{d\mathcal{N}(e^{-t/2} x_0, (1 - e^{-t})\mathcal{C})}{d\mathcal{N}(0, (1 - e^{-t})\mathcal{C})}(x) \\
&= \mathbb{E}(X_0^\mu | Y = y, X_t^\mu = x),
\end{aligned}
$$

for $(x, y) \in H \in \mathbb{R}^m$ a.e. in $\mathscr{L}(X_t^\mu, Y)$, where the last equality follows from the proof of Theorem 2 in Pidstrigach et al. (2024) and that $\mathcal{N}(e^{-t/2} x_0, (1 - e^{-t})\mathcal{C})$ is the transition kernel of the forward SDE in Equation 4.

**(ii):** We repeat the aforementioned argument from Pidstrigach et al. (2024) adapted to our case. The joint distribution of $\widetilde{X}_t^\mu, \widetilde{X}_0^\mu$ is given by $\tilde{n}(x_0, x)(\mathcal{N}(0, \Sigma_t))(dx) \otimes \mu(dx_0)$. Indeed, for any $\mathcal{A} \in \sigma(\widetilde{X}_0^\mu), \mathcal{B} \in \sigma(\widetilde{X}_t^\mu)$,

$$
\begin{aligned}
\iint_{\mathcal{A} \times \mathcal{B}} \tilde{n}(x_0, x) \mathcal{N}(0, \Sigma_t)(dx) \mu(dx_0) &= \iint_{\mathcal{A} \times \mathcal{B}} \frac{d\mathcal{N}(x_0, \Sigma_t)}{d\mathcal{N}(0, \Sigma_t)}(x) \mathcal{N}(0, \Sigma_t)(dx) \mu(dx_0) \\
&= \iint_{\mathcal{A} \times \mathcal{B}} \mathcal{N}(x_0, \Sigma_t)(dx) \mu(dx_0) \\
&= \mathbb{P}(\widetilde{X}_0^\mu \in \mathcal{A}, \widetilde{X}_t^\mu \in \mathcal{B}),
\end{aligned}
$$

where it is used that $\mathcal{N}(x_0, \Sigma_t)$ is the forward transition kernel of the process $\widetilde{X}^\mu$. We show that

$$f(x) = \frac{\int x_0 \tilde{n}_t(x_0, x) \mu(dx_0)}{\int \tilde{n}_t(x_0, x) \mu(dx_0)}$$

is a version of the conditional expectation $\mathbb{E}(\widetilde{X}_0^\mu | \widetilde{X}_t^\mu = x)$. Let $\mathbb{P}_t$ be the law of $\widetilde{X}_t^\mu$, that is,

$$\mathbb{P}_t(A) = \mathbb{P}(\widetilde{X}_t^\mu \in A), \quad A \in \sigma(\widetilde{X}_t^\mu).$$

The function $f(x)$ is $\sigma(\widetilde{X}_t^\mu)$-measurable by Fubini's theorem and for any $\mathcal{A} \in \sigma(\widetilde{X}_t^\mu)$,

$$
\begin{aligned}
\mathbb{E}_{x \sim \mathscr{L}(X_t^\mu)} (1_{\mathcal{A}} f(x)) &= \int_{\mathcal{A}} \frac{\int_H x_0 \tilde{n}_t(x_0, x) \mu(dx_0)}{\int_H \tilde{n}_t(x_0, x) \mu(dx_0)} d\mathbb{P}_t(x) \\
&= \iint_{H \times \mathcal{A}} \frac{\int_H x_0 \tilde{n}_t(x_0, x) \mu(dx_0)}{\int_H \tilde{n}_t(x_0, x) \mu(dx_0)} \tilde{n}(\tilde{x}_0, x_t) \mathcal{N}(0, \Sigma_t)(dx) \mu(d\tilde{x}_0) \\
&= \iint_{\mathcal{A} \times H} x_0 \tilde{n}_t(x_0, x) \mu(dx_0) \frac{\int_H \tilde{n}(\tilde{x}_0, x) \mu(d\tilde{x}_0)}{\int_H \tilde{n}_t(x_0, x) \mu(dx_0)} \mathcal{N}(0, \Sigma_t)(dx) \\
&= \iint_{\mathcal{A} \times H} x_0 \tilde{n}_t(x_0, x) \mu(dx_0) \mathcal{N}(0, \Sigma_t)(dx) = \mathbb{E}_{x_0 \sim \widetilde{X}_0^\mu} (1_{\mathcal{A}} x_0).
\end{aligned}
$$

The above properties define the conditional expectation and we can conclude the proof. $\square$

**Lemma B.6.** *We have that*

(i) $\Sigma_t(H) = \mathcal{C}(H)$; moreover,

$$\Sigma_t^{-1}\big|_{\Sigma_t(H)} = (A^*\Gamma^{-1}A + (e^t - 1)^{-1}\mathcal{C}^{-1})\big|_{\mathcal{C}(H)}. \tag{26}$$

(ii) $H_{(e^t-1)\mathcal{C}} = H_{\Sigma_t}$.

(iii) The measures $\mathcal{N}(0, (e^t - 1)\mathcal{C})$ and $\mathcal{N}(0, \Sigma_t)$ are equivalent.

(iv) For $x_0 \in H_{(e^t-1)\mathcal{C}}$, we have

$$\langle x_0, \cdot \rangle_{H_{\Sigma_t}} = \langle x_0, A^*\Gamma^{-1}A\cdot \rangle_H + \langle x_0, \cdot \rangle_{H_{(e^t-1)\mathcal{C}}} \tag{27}$$

in $\mathcal{N}(0, \Sigma_t)$-a.e.

*Proof.* For the purpose of this proof, we abbreviate

$$\widetilde{\mathcal{C}}_t = (e^t - 1)\mathcal{C}. \tag{28}$$

**(i):** We first show that $\widetilde{\mathcal{C}}_t(H) = \Sigma_t(H)$. Note that we can write $\Sigma_t$ as

$$\Sigma_t = \widetilde{\mathcal{C}}_t \left( I - \widetilde{\mathcal{C}}_t A^* C_t^{-1} A \right),$$

where the last factor on the right-hand side is invertible (see Lemma B.1 (b)) proving that ranges of $\Sigma_t$ and $\widetilde{\mathcal{C}}_t$ coincide. To prove Equation 26, let $z \in \Sigma_t(H) = \mathcal{C}(H)$, then for some $x \in H$,

$$z = \Sigma_t x = \widetilde{\mathcal{C}}_t x - \widetilde{\mathcal{C}}_t A^* C_t^{-1} A \widetilde{\mathcal{C}}_t x.$$

We have

$$\left( A^*\Gamma^{-1}A + \widetilde{\mathcal{C}}_t^{-1} \right) z = A^*\Gamma^{-1}A\widetilde{\mathcal{C}}_t x - A^*\Gamma^{-1}A\widetilde{\mathcal{C}}_t A^* C_t^{-1} A \widetilde{\mathcal{C}}_t x + x - A^* C_t^{-1} A \widetilde{\mathcal{C}}_t x$$

$$= x + A^*\Gamma^{-1}A\widetilde{\mathcal{C}}_t x - A^*\Gamma^{-1} \left( \Gamma + A\widetilde{\mathcal{C}}_t A^* \right) C_t^{-1} A \widetilde{\mathcal{C}}_t x = x,$$

since $\Gamma + A\widetilde{\mathcal{C}}_t A^* = C_t$. This shows that

$$\Sigma_t^{-1} z = x = \left( A^*\Gamma^{-1}A + \widetilde{\mathcal{C}}_t^{-1} \right) z.$$

**(ii):** Notice first that we can write

$$H_{\widetilde{\mathcal{C}}_t} = \mathrm{cl}^{\|\cdot\|_{H_{\widetilde{\mathcal{C}}_t}}} (\widetilde{\mathcal{C}}_t(H)), \quad H_{\Sigma_t} = \mathrm{cl}^{\|\cdot\|_{H_{\Sigma_t}}} (\Sigma_t(H)),$$

where $\mathrm{cl}^{\|\cdot\|_{H_1}}(H_2)$ denotes the closure of $H_2 \subset H_1$ w.r.t $\|\cdot\|_{H_1}$. Let us prove that the norms $\|\cdot\|_{H_{\widetilde{\mathcal{C}}_t}}$ and $\|\cdot\|_{H_{\Sigma_t}}$ are equivalent on $\Sigma_t(H) = \widetilde{\mathcal{C}}_t(H)$ as, together with (i), this will prove the statement.

To this end, let $x \in \Sigma_t(H) = \widetilde{\mathcal{C}}_t(H)$ and apply $(i)$ and Cauchy–Schwarz inequality to obtain

$$\|x\|_{H_{\widetilde{\mathcal{C}}_t}}^2 \leq \|x\|_{H_{\Sigma_t}}^2 = \langle x, A^*\Gamma^{-1}Ax \rangle_H + \|x\|_{H_{\widetilde{\mathcal{C}}_t}}^2$$

$$\leq \left\| \widetilde{\mathcal{C}}_t^{1/2} A^*\Gamma^{-1} A \widetilde{\mathcal{C}}_t^{1/2} \right\|_{\mathcal{L}(H,H)} \|x\|_{H_{\widetilde{\mathcal{C}}_t}}^2 + \|x\|_{H_{\widetilde{\mathcal{C}}_t}}^2$$

$$= \left( 1 + \left\| \widetilde{\mathcal{C}}_t^{1/2} A^*\Gamma^{-1} A \widetilde{\mathcal{C}}_t^{1/2} \right\|_{\mathcal{L}(H,H)} \right) \|x\|_{H_{\widetilde{\mathcal{C}}_t}}^2,$$

where the operator norm $\left\| \widetilde{\mathcal{C}}_t^{1/2} A^*\Gamma^{-1} A \widetilde{\mathcal{C}}_t^{1/2} \right\|_{\mathcal{L}(H,H)}$ is finite.

**(iii):** From $(ii)$, the covariance operators of $\mu$ and $\nu$ have the same Cameron–Martin space, by the Feldman–Hajek theorem we need to show that $\widetilde{\mathcal{C}}_t^{-1/2}\Sigma_t\widetilde{\mathcal{C}}_t^{-1/2} - I$ is Hilbert-Schmidt. To this end, note that

$$\widetilde{\mathcal{C}}_t^{-1/2}\Sigma_t\widetilde{\mathcal{C}}_t^{-1/2} - I = -\widetilde{\mathcal{C}}_t^{1/2}A^*C_t^{-1}A\widetilde{\mathcal{C}}_t^{1/2} := B.$$

The operator $B^2$ is of trace class, since

$$
\begin{aligned}
\mathrm{tr}_H(B^2) &= \mathrm{tr}_H\left(\widetilde{\mathcal{C}}_t^{1/2}A^*C_t^{-1}A\widetilde{\mathcal{C}}_tA^*C_t^{-1}A\widetilde{\mathcal{C}}_t^{1/2}\right) \\
&\leq \left\|A^*C_t^{-1}A\widetilde{\mathcal{C}}_tA^*C_t^{-1}A\right\|_{\mathcal{L}(H,H)}\mathrm{tr}_H(\widetilde{\mathcal{C}}_t) < \infty,
\end{aligned}
$$

proving the claim.

**(iv):** Let $x_0 \in \Sigma_t(H)$. By $(i)$, we have

$$
\begin{aligned}
\langle x_0,\cdot\rangle_{H_{\Sigma_t}} &= \langle \Sigma_t^{-1}x_0,\cdot\rangle_H \\
&= \langle A^*\Gamma^{-1}Ax_0,\cdot\rangle_H + \langle \widetilde{\mathcal{C}}_t^{-1}x_0,\cdot\rangle_H \qquad\qquad (29) \\
&= \langle A^*\Gamma^{-1}Ax_0,\cdot\rangle_H + \langle x_0,\cdot\rangle_{H_{\widetilde{\mathcal{C}}_t}} \quad \mathcal{N}(0,\Sigma_t)\text{-a.e.}
\end{aligned}
$$

Now since $\Sigma_t(H) = \mathcal{C}(H)$ is dense in $H_{\Sigma_t} = H_{\widetilde{\mathcal{C}}_t}$, the identity above can be uniquely extended to $H_{\Sigma_t}$ by the white noise mapping, see (Da Prato, 2006, p. 23). This completes the proof. □

**Corollary B.7.** *We have*

$$e^{t/2}\langle x_0, X_t^\mu\rangle_{H_{\widetilde{\mathcal{C}}_t}} + \langle x_0, A^*\Gamma^{-1}Y\rangle_H = \langle x_0, m_t(X_t^\mu,Y)\rangle_{H_{\Sigma_t}} \qquad\qquad (30)$$

*in distribution.*

*Proof.* Combining Lemmas B.6 $(iv)$ and B.4, it follows that

$$\langle x_0, m_t(X_t^\mu,Y)\rangle_{H_{\Sigma_t}} = \langle x_0, m_t(X_t^\mu,Y)\rangle_{H_{\widetilde{\mathcal{C}}_t}} + \langle x_0, A^*\Gamma^{-1}Am_t(X_t^\mu,Y)\rangle_H$$

in distribution. Therefore, we obtain

$$
\begin{aligned}
&\langle x_0, m_t(X_t^\mu,Y)\rangle_{H_{\Sigma_t}} \\
&= \langle x_0, m_t(X_t^\mu,Y)\rangle_{H_{\widetilde{\mathcal{C}}_t}} + \langle x_0, A^*\Gamma^{-1}Am_t(X_t^\mu,Y)\rangle_H \\
&= e^{t/2}\langle x_0, X_t^\mu\rangle_{H_{\widetilde{\mathcal{C}}_t}} + \langle x_0, \widetilde{\mathcal{C}}_tA^*C_t^{-1}(Y - e^{t/2}AX_t^\mu)\rangle_{H_{\widetilde{\mathcal{C}}_t}} + e^{t/2}\langle x_0, A^*\Gamma^{-1}AX_t^\mu\rangle_H \\
&\quad + \langle x_0, (e^t-1)A^*\Gamma^{-1}A\mathcal{C}A^*C_t^{-1}(Y - e^{t/2}AX_t^\mu)\rangle_H \\
&= e^{t/2}\langle x_0, X_t^\mu\rangle_{H_{\widetilde{\mathcal{C}}_t}} + e^{t/2}\langle x_0, A^*\Gamma^{-1}AX_t^\mu\rangle_H + \langle x_0, A^*C_t^{-1}(Y - e^{t/2}AX_t^\mu)\rangle_H \\
&\quad + (e^t-1)\langle x_0, A^*\Gamma^{-1}A\mathcal{C}A^*C_t^{-1}(Y - e^{t/2}AX_t^\mu)\rangle_H \\
&= e^{t/2}\langle x_0, X_t^\mu\rangle_{H_{\widetilde{\mathcal{C}}_t}} + e^{t/2}\langle x_0, A^*\Gamma^{-1}AX_t^\mu\rangle_H + \langle x_0, A^*\Gamma^{-1}(Y - e^{t/2}X_t^\mu)\rangle_H \\
&= e^{t/2}\langle x_0, X_t^\mu\rangle_{H_{\widetilde{\mathcal{C}}_t}} + \langle x_0, A^*\Gamma^{-1}Y\rangle_H
\end{aligned}
$$

in distribution, where we have used that

$$A^*C_t^{-1} + (e^t-1)A^*\Gamma^{-1}A\mathcal{C}A^*C_t^{-1} = A^*\Gamma^{-1}\underbrace{\left(\Gamma + (e^t-1)A\mathcal{C}A^*\right)}_{=C_t}C_t^{-1} = A^*\Gamma^{-1}.$$

This completes the proof. □

*Proof of Proposition B.3.* Let $x_0 \in H_{\widetilde{\mathcal{C}}_t} \subset H$ where $\widetilde{\mathcal{C}}_t = (e^t - 1)\mathcal{C}$. By virtue of Lemma B.5, we will prove that

$$\frac{\int_H x_0 n_t(x_0, x) \mu^y(dx_0)}{\int_H n_t(x_0, x) \mu^y(dx_0)} = \frac{\int_H x_0 \tilde{n}_t(x_0, m_t(x, y)) \mu(dx_0)}{\int_H \tilde{n}_t(x_0, m_t(x, y)) \mu(dx_0)}, \tag{31}$$

for $(x, y) \in H \in \mathbb{R}^m$ a.e. in $\mathscr{L}(X_t^\mu, Y)$. Notice that it suffices to show that

$$n_t(x_0, x) \frac{d\mu^y}{d\mu}(x_0) = \frac{1}{\widetilde{Z}(y)} \tilde{n}_t(x_0, m_t(x, y)), \tag{32}$$

for $x_0 \in H$ $\mu$-a.e. and for $(x, y) \in H \in \mathbb{R}^m$ a.e. in $\mathscr{L}(X_t^\mu, Y)$, where $\widetilde{Z}(y)$ does not depend on $x_0$, hence will be canceled out in Equation 31. By Bayes' theorem Equation 7,

$$\frac{d\mu^y}{d\mu}(\cdot) = \frac{1}{Z(y)} \exp\left(-\frac{1}{2} \|A \cdot -y\|_\Gamma^2\right) \quad \text{in } L^1(\mu),$$

Let us now write

$$-\frac{1}{2} \|Ax_0 - y\|_\Gamma^2 = \langle x_0, A^*\Gamma^{-1}y \rangle_H - \frac{1}{2} \langle x_0, A^*\Gamma^{-1}Ax_0 \rangle_H - \frac{1}{2} \|y\|_\Gamma^2$$

and set $\widetilde{Z}(y) = Z(y) \exp\left(\frac{1}{2} \|y\|_\Gamma^2\right)$. For $x_0 \in H_{\widetilde{\mathcal{C}}_t}$,

$$\begin{aligned}
&n_t(x_0, X_t^\mu) \exp\left(\langle x_0, A^*\Gamma^{-1}Y \rangle_H - \frac{1}{2} \langle x_0, A^*\Gamma^{-1}Ax_0 \rangle_H - \frac{1}{2} \|Y\|_\Gamma^2\right) \\
&= \frac{1}{\widetilde{Z}(Y)} \exp\left(e^{t/2} \langle x_0, X_t^\mu \rangle_{H_{\widetilde{\mathcal{C}}_t}} - \frac{1}{2} \|x_0\|_{H_{\widetilde{\mathcal{C}}_t}}^2 + \langle x_0, A^*\Gamma^{-1}Y \rangle_H - \frac{1}{2} \langle x_0, A^*\Gamma^{-1}Ax_0 \rangle_H\right) \\
&= \frac{1}{\widetilde{Z}(Y)} \exp\left(e^{t/2} \langle x_0, X_t^\mu \rangle_{H_{\widetilde{\mathcal{C}}_t}} + \langle x_0, A^*\Gamma^{-1}Y \rangle_H - \frac{1}{2} \|x_0\|_{H_{\widetilde{\mathcal{C}}_t}}^2 - \frac{1}{2} \langle x_0, A^*\Gamma^{-1}Ax_0 \rangle_H\right) \\
&= \frac{1}{\widetilde{Z}(Y)} \exp\left(\langle x_0, m_t(X_t^\mu, Y) \rangle_{H_{\Sigma_t}} - \frac{1}{2} \|x_0\|_{H_{\Sigma_t}}^2\right),
\end{aligned} \tag{33}$$

in distribution, where we have used Corollary B.7 and the identity

$$\|x_0\|_{H_{\widetilde{\mathcal{C}}_t}}^2 + \langle x_0, A^*\Gamma^{-1}Ax_0 \rangle_H = \|x_0\|_{H_{\Sigma_t}}^2, \tag{34}$$

which follows from Lemma B.6 $(iv)$.

Now by Lemma B.6 $(ii)$, $x_0 \in H_{\Sigma_t}$ and thus the Cameron–Martin theorem gives

$$\exp\left(\langle x_0, m_t(X_t^\mu, Y) \rangle_{H_{\Sigma_t}} - \frac{1}{2} \|x_0\|_{H_{\Sigma_t}}^2\right) = \tilde{n}_t(x_0, m_t(X_t^\mu, Y))$$

in distribution, which together with Equation 33 lead to Equation 32. Therefore, Equation 31 holds true, which completes the proof. $\qquad\square$

### B.3 Gaussian example

*Proof for Lemma 3.8.* Let $\widetilde{\mathcal{C}}$ be a covariance operator such that $\mu\left(\widetilde{\mathcal{C}}^{1/2}(H)\right) = 1$. As $S_0^{1/2}(H)$ is the intersection of all linear subspaces of full measure under $\mu$ (see Prop. 4.45 in Hairer, 2023), it holds $S_0^{1/2}(H) \subset \widetilde{\mathcal{C}}^{1/2}(H)$.

We now find another covariance operator $\mathcal{C}$ such that the score function $\tilde{s}$ corresponding to $\mathcal{C}$ is bounded linear. Let $\mathcal{C}$ be such that $\mathcal{C}^{1/2}(X) \subset S_0^{1/2}(X)$ for any linear subset $X$ of $H$. This implies

$$S_0(H) \subset \mathcal{C}(H) = \Sigma_t(H). \tag{35}$$

We identify the law of $\widetilde{X}_t^\mu$ by using the relation

$$\widetilde{X}_t^\mu | x_0 \sim \mathcal{N}(x_0, \Sigma_t),$$

where $x_0$ is a realisation of the prior $\mu = \mathcal{N}(0, S_0)$. Hence

$$\widetilde{X}_t^\mu \sim \mathcal{N}(0, \Sigma_t + S_0).$$

By the reasoning of Lemma 4.4 in Hairer et al. (2006) it holds that

$$\widetilde{X}_0^\mu | \widetilde{X}_t^\mu \sim \mathcal{N}(m', C')$$

with some covariance operator $C'$ and

$$m' \;\; = \;\; \Sigma_t(\Sigma_t + S_0)^{-1} \widetilde{X}_t^\mu.$$

Note that in Lemma 4.4 of Hairer et al. (2006), the operator $\Sigma_t(\Sigma_t + S_0)^{-1}$ is defined as a measurable extension of the bounded map

$$A : (\Sigma_t + S_0)^{1/2}(H) \to H, \qquad x \mapsto \Sigma_t(\Sigma_t + S_0)^{-1}x$$

to the whole space $H$ as per Theorem II.3.3 in Daleckij (1991). In our case, it is possible to give an explicit formula for a possible extension using the inclusion Equation 35. We define

$$\Sigma_t(\Sigma_t + S_0)^{-1} \;\; := \;\; \left((\Sigma_t + S_0)^{-1}\Sigma_t\right)^*$$
$$= \;\; \left((I + \Sigma_t^{-1}S_0)^{-1}\Sigma_t^{-1}S_0\right)^*.$$

This map coincides with $A$ on $(\Sigma_t + S_0)^{1/2}(H)$ and as we will now show, it is defined even on the whole space $H$. The operator $\Sigma_t : H \to \Sigma_t(H)$ is bounded and invertible. Hence also $\Sigma_t^{-1} : \Sigma_t(H) \to H$ is bounded. By Equation 35 the map $\Sigma_t^{-1}S_0$ is well defined and bounded. We can now identify the score function $\tilde{s}$ by using the previous equality for the conditional expectation

$$\tilde{s}(z, t; \mu) \;\; = \;\; -\left(z - \mathbb{E}(\widetilde{X}_0^\mu | \widetilde{X}_t^\mu = z)\right)$$
$$= \;\; -z + S_0(\Sigma_t + S_0)^{-1}z$$
$$= \;\; \left[S_0(\Sigma_t + S_0)^{-1} - I\right] z.$$
$$= \;\; -\Sigma_t(\Sigma_t + S_0)^{-1}z.$$

This yields the claim. $\qquad\square$

### B.4 Conditional score matching

Recall that $m_t(x, y) = e^{t/2}x + (e^t - 1)\mathcal{C}A^*C_t^{-1}(y - e^{t/2}Ax)$ and $\xi_t(x, y) := \mathcal{C}A^*\Gamma^{-1}y + \lambda(t)x$ for $x \in H$ and $y \in \mathbb{R}^m$. Moreover, $R_t := (e^t - 1)I + (e^t - 1)^2\mathcal{C}A^*C_t^{-1}A : H \to H$.

**Lemma B.8.** *The operator $R_t : H \to H$ is bijective and the identity*

$$m_t(x, y) = R_t\xi_t(x, y).$$

*holds for all $x \in H$ and $y \in \mathbb{R}^m$.*

*Proof.* First note that bijectivity is implied by Lemma B.1 (b). By direct computation,

$$R_t\xi_t(x, y) \;\; = \;\; R_t\left(\mathcal{C}A^*\Gamma^{-1}y + \lambda(t)x\right)$$
$$= \;\; (e^t - 1)\mathcal{C}A^*\Gamma^{-1}y - (e^t - 1)^2\mathcal{C}A^*C_t^{-1}A\mathcal{C}A^*\Gamma^{-1}y$$
$$+ e^{t/2}x - e^{t/2}(e^t - 1)\mathcal{C}A^*C_t^{-1}Ax$$
$$= \;\; e^{t/2}x - e^{t/2}(e^t - 1)\mathcal{C}A^*C_t^{-1}Ax$$
$$+ (e^t - 1)\mathcal{C}A^*(C_t^{-1}\underbrace{[(e^t - 1)A\mathcal{C}A^* + \Gamma]}_{=C_t} + (e^t - 1)C_t^{-1}A\mathcal{C}A^*)\Gamma^{-1}y$$
$$= \;\; e^{t/2}x + (e^t - 1)\mathcal{C}A^*C_t^{-1}\left(y - e^{t/2}Ax\right)$$
$$= \;\; m_t(x, y).$$

This completes the proof. □

*Proof of Lemma 3.10.* We replicate the arguments of Baldassari et al. (2024a) adapted to our setting. First, observe that

$$\left\|r_\theta(R_t^{-1}\tilde{x}_t,t;\mu) - \tilde{s}(\tilde{x}_t,t;\mu) - \tilde{x}_t\right\|_H^2$$
$$= \left\|r_\theta(R_t^{-1}\tilde{x}_t,t;\mu)\right\|_H^2 + \left\|\tilde{s}(\tilde{x}_t,t;\mu) - \tilde{x}_t\right\|_H^2 - 2\langle r_\theta(R_t^{-1}\tilde{x}_t,t;\mu), \tilde{s}(\tilde{x}_t,t;\mu) - \tilde{x}_t\rangle_H.$$

It holds by definition Equation 14

$$\mathbb{E}_{\tilde{x}_t\sim\mathscr{L}(\widetilde{X}_t^\mu)}\langle r_\theta(R_t^{-1}\tilde{x}_t,t;\mu), \tilde{s}(\tilde{x}_t,t;\mu) - \tilde{x}_t\rangle_H$$
$$= -\mathbb{E}_{\tilde{x}_t\sim\mathscr{L}(\widetilde{X}_t^\mu)}\left\langle r_\theta(R_t^{-1}\tilde{x}_t,t;\mu), \mathbb{E}_{\tilde{x}_0\sim\mathscr{L}(\widetilde{X}_0^\mu|\tilde{x}_t)}(\tilde{x}_t - \tilde{x}_0) - \tilde{x}_t\right\rangle_H$$
$$= \mathbb{E}_{\tilde{x}_0\sim\mathscr{L}(\widetilde{X}_0^\mu)}\mathbb{E}_{\tilde{x}_t\sim\mathscr{L}(\widetilde{X}_t^\mu|\tilde{x}_0)}\langle r_\theta(R_t^{-1}\tilde{x}_t,t;\mu), \tilde{x}_0\rangle_H.$$

Hence it holds

$$\lambda(t)^2\mathbb{E}_{\tilde{x}_t\sim\mathscr{L}(\widetilde{X}_t^\mu)}\left\|r_\theta(R_t^{-1}\tilde{x}_t,t;\mu) - \tilde{s}(\tilde{x}_t,t;\mu) + \tilde{x}_t\right\|_H^2$$
$$= V'(t) + \lambda(t)^2\mathbb{E}_{\tilde{x}_0\sim\mathscr{L}(\widetilde{X}_0^\mu)}\mathbb{E}_{\tilde{x}_t\sim\mathscr{L}(\widetilde{X}_t^\mu|\tilde{x}_0)}\left\|r_\theta(R_t^{-1}\tilde{x}_t,t;\mu) - \tilde{x}_0\right\|_H^2$$

with

$$V'(t) = \lambda(t)^2\mathbb{E}_{\tilde{x}_t\sim\mathscr{L}(\widetilde{X}_t^\mu)}\left\|\tilde{s}(\tilde{x}_t,t,\mu) - \tilde{x}_t\right\|_H^2 - \lambda(t)^2\mathbb{E}_{\tilde{x}_0\sim\mathscr{L}(\widetilde{X}_0^\mu)}\left\|\tilde{x}_0\right\|_H^2. \tag{36}$$

To conclude, we add expectation with respect to $t\sim[\delta,T]$. Note that by Assumption 3.9 the first term on the rhs of Equation 36 is uniformly bounded in $t$ and by elementary calculations

$$\mathbb{E}_{\tilde{x}_0\sim\mathscr{L}(\widetilde{X}_0^\mu)}\lambda(t)^2\left\|\tilde{x}_0\right\|_H^2 \leq \lambda(\delta)^2\mathbb{E}_{x\sim\mu}\left\|x\right\|_H,$$

such that

$$V := \mathbb{E}_{t\sim[\delta,T]}V'(t) \leq \frac{1}{T-\delta}\sup_{t\in[\delta,T]}V'(t) < \infty.$$

This concludes the proof.

□

## C  Proofs for Section 4

*Proof of Theorem 4.1.* Below, we use the notation $f \lesssim g$, if $f(x) \leq Cg(x)$ for all $x$ with some universal constant $C > 0$. Recall that true solution $W_t^{\mu^y}$ of the time-reversed denoising process and corresponding approximative solution process $V_t^{\mu^y}$ satisfy

$$dW_t^{\mu^y} = \left(-\frac{1}{2}W_t^{\mu^y} - s(W_t^{\mu^y}, T-t;\mu^y)\right)dt + \mathcal{C}^{1/2}dB_t,$$

$$dV_t^{\mu^y} = -\frac{1}{2}V_{\lfloor t\rfloor}^{\mu^y} - \lambda(T-\lfloor t\rfloor)\left[r_\theta\left(\xi_{T-\lfloor t\rfloor}(V_{\lfloor t\rfloor}^{\mu^y},y), T-\lfloor t\rfloor;\mu\right) - e^{(T-\lfloor t\rfloor)/2}V_{\lfloor t\rfloor}^{\mu^y}\right]dt + \mathcal{C}^{1/2}dB_t.$$

In what follows, we abbreviate the expectation $\mathbb{E}_{y\sim\pi_y}\mathbb{E}_{(w_T,v_{T-\delta})\sim\mathscr{L}(W_T^{\mu^y},V_{T-\delta}^{\mu^y})}$ as $\mathbb{E}$, unless otherwise specified.

**Decomposition of the error.** Let us first consider the difference

$$\mathbb{E}\left\|w_t - v_t\right\|_H^2 = \mathbb{E}\left\|w_t - w_0 - (v_t - v_0) + (w_0 - v_0)\right\|_H^2$$
$$\lesssim \mathbb{E}\left\|w_t - w_0 - (v_t - v_0)\right\|_H^2 + \varepsilon_{INIT},$$

where $\varepsilon_{INIT} := \mathbb{E}\left\| w_0 - v_0 \right\|_H^2$. Now we decompose the difference into three terms as follows:

$$w_t - w_0 - (v_t - v_0) = \int_0^t \left[ \mathcal{I}_1(\tau) + \mathcal{I}_2(\tau) + \mathcal{I}_3(\tau) \right] d\tau,$$

where we have

$$
\begin{aligned}
\mathcal{I}_1(\tau) &= -\frac{1}{2}\left( w_\tau - w_{\lfloor \tau \rfloor} \right) + s(w_\tau, T - \tau; \mu^y) - s(w_{\lfloor \tau \rfloor}, T - \lfloor \tau \rfloor; \mu^y), \\
\mathcal{I}_2(\tau) &= s(w_{\lfloor \tau \rfloor}, T - \lfloor \tau \rfloor; \mu^y) - \lambda(T - \lfloor \tau \rfloor)\left( r_\theta(\xi_{T-\lfloor \tau \rfloor}(w_{\lfloor \tau \rfloor}, y), T - \lfloor \tau \rfloor; \mu) - e^{(T-\lfloor \tau \rfloor)/2} w_{\lfloor \tau \rfloor} \right) \quad \text{and} \\
\mathcal{I}_3(\tau) &= \lambda(T - \lfloor \tau \rfloor)\left\{ r_\theta(\xi_{T-\lfloor \tau \rfloor}(w_{\lfloor \tau \rfloor}, y), T - \lfloor \tau \rfloor; \mu) - r_\theta(\xi_{T-\lfloor \tau \rfloor}(v_{\lfloor \tau \rfloor}, y), T - \lfloor \tau \rfloor; \mu) \right\} \\
&\quad - (\lambda(T - \lfloor \tau \rfloor)e^{(T-\lfloor \tau \rfloor)/2} + 1)\left( w_{\lfloor \tau \rfloor} - v_{\lfloor \tau \rfloor} \right).
\end{aligned}
$$

**Bound for $\varepsilon_{INIT}$.** Recall that $v_0 = (1 - e^{-T})Z$ and $w_0 = (1 - e^{-T})Z + e^{-T/2}X_0$, where $Z \sim \mathcal{N}(0, \mathcal{C})$ and $X_0 \sim \mu^y$. It directly follows that

$$\varepsilon_{INIT} \leq e^{-T}\mathbb{E}_{y \sim \pi_y}\mathbb{E}_{x \sim \mu^y}\left\| x \right\|_H^2 = e^{-T}\mathbb{E}_{x \sim \mu}\left\| x \right\|_H^2.$$

where we applied marginalization of the joint distribution.

**Contribution from $\mathcal{I}_1$:** We observe that

$$
\begin{aligned}
\varepsilon_{NUM} &:= \mathbb{E}\int_0^{T-\delta}\left\| \mathcal{I}_1(\tau) \right\|_H^2 d\tau \\
&= \mathbb{E}\int_0^{T-\delta}\left\| -\frac{1}{2}\left( w_\tau - w_{\lfloor \tau \rfloor} \right) + s(w_\tau, T - \tau; \mu^y) - s(w_{\lfloor \tau \rfloor}, T - \lfloor \tau \rfloor; \mu^y) \right\|_H^2 d\tau \\
&\lesssim \mathbb{E}\int_0^{T-\delta}\left\| w_\tau - w_{\lfloor \tau \rfloor} \right\|_H^2 d\tau + \mathbb{E}\int_0^{T-\delta}\left\| s(w_\tau, T - \tau; \mu^y) - s(w_{\lfloor \tau \rfloor}, T - \lfloor \tau \rfloor; \mu^y) \right\|_H^2 d\tau.
\end{aligned}
$$

By Lemma 2 in Pidstrigach et al. (2024), it holds that

$$s(X_t^{\mu^y}, t; \mu^y) = e^{(t-\tau)/2}\mathbb{E}\left( s(X_\tau^{\mu^y}, \tau; \mu^y)|X_t^{\mu^y} \right), \quad 0 < t \leq \tau \leq T,$$

for $\pi_y$-a.e. $y \in \mathbb{R}^m$. Therefore, we deduce by Lemma 11 in Chen et al. (2023a) that for the time-reversed process it holds that

$$
\begin{aligned}
\mathbb{E}&\left\| s(w_\tau, T - \tau; \mu^y) - s(w_{\lfloor \tau \rfloor}, T - \lfloor \tau \rfloor; \mu^y) \right\|_H^2 \\
&\leq 4\mathbb{E}\left\| s(w_\tau, T - \tau; \mu^y) - s(e^{(\tau-\lfloor \tau \rfloor)/2}w_{\lfloor \tau \rfloor}, T - \tau; \mu^y) \right\|_H^2 \\
&\quad + 2\left( 1 - e^{\tau - \lfloor \tau \rfloor} \right)^2 \mathbb{E}\left\| s(w_\tau, T - \tau; \mu^y) \right\|_H^2 \\
&\leq 4L_s^2(T - \tau)\mathbb{E}\left\| w_\tau - e^{(t-\lfloor t \rfloor)/2}w_{\lfloor \tau \rfloor} \right\|_H^2 + 2(1 - e^{\tau - \lfloor \tau \rfloor})^2\mathbb{E}\left\| s(w_\tau, T - \tau; \mu^y) \right\|_H^2,
\end{aligned}
$$

where we used the Lipschitz continuity of $s$. We note that

$$w_s|w_t \sim \mathcal{N}\left( e^{-(t-s)/2}w_t, (1 - e^{-(t-s)})\mathcal{C} \right), \tag{37}$$

for $T \geq t \geq s \geq 0$, see Itô (1984). It immediately follows that

$$\mathbb{E}\left\| w_\tau - e^{(\tau-\lfloor \tau \rfloor)/2}w_{\lfloor \tau \rfloor} \right\|_H^2 = (1 - e^{-(\tau-\lfloor \tau \rfloor)})\mathrm{tr}_H(\mathcal{C}),$$

and, consequently,

$$\mathbb{E}\left\|w_\tau - w_{\lfloor\tau\rfloor}\right\|_H^2 \lesssim \mathbb{E}\left\|w_\tau - e^{(\tau-\lfloor\tau\rfloor)/2}w_{\lfloor\tau\rfloor}\right\|_H^2 + (1-e^{(\tau-\lfloor\tau\rfloor)/2})^2\mathbb{E}\left\|w_{\lfloor\tau\rfloor}\right\|_H^2$$

$$\lesssim (1-e^{-(\tau-\lfloor\tau\rfloor)})\mathrm{tr}_H(\mathcal{C})$$
$$+ (1-e^{(\tau-\lfloor\tau\rfloor)/2})^2\left(\mathbb{E}_{y\sim\pi_y}\mathbb{E}_{x\sim\mu^y}\|x\|_H^2 + \left((1-e^{-(T-\lfloor\tau\rfloor)})\right)\mathrm{tr}_H(\mathcal{C})\right)$$

$$\lesssim (1-e^{-\Delta t})\mathrm{tr}_H(\mathcal{C}) + (1-e^{\Delta t/2})^2\left(\mathbb{E}_{x\sim\mu}\|x\|_H^2 + \mathrm{tr}_H(\mathcal{C})\right).$$

Combining the arguments yields

$$\varepsilon_{NUM} \lesssim (T-\delta)\left((1-e^{-\Delta t})\mathrm{tr}_H(\mathcal{C}) + (1-e^{\Delta t/2})^2\left(\mathbb{E}_{x\sim\mu}\|x\|_H^2 + \mathrm{tr}_H(\mathcal{C})\right)\right)$$
$$+ L_s^2(T-\tau)(1-e^{-\Delta t})\mathrm{tr}_H(\mathcal{C}) + (1-e^{\Delta t})^2\mathbb{E}\|s(w_\tau, T-\tau; \mu^y)\|_H^2.$$

Since the last expectation and $\mathbb{E}_{x\sim\mu}\|x\|_H^2$ are bounded by assumption, and note that $1-e^{-\Delta t} = \mathcal{O}(\Delta t)$, $(1-e^{\Delta t/2})^2 = \mathcal{O}(\Delta t)$, we obtain the required upper bound.

**Contribution from $\mathcal{I}_2$:** Applying Theorem 3.7, we obtain

$$\mathcal{I}_2(\tau) = \lambda(T-\lfloor\tau\rfloor)\left\{\tilde{s}(R_{T-\lfloor\tau\rfloor}\xi(w_{\lfloor\tau\rfloor}, y), T-\lfloor\tau\rfloor; \mu) + R_{T-\lfloor\tau\rfloor}\xi(w_{\lfloor\tau\rfloor}, y)\right.$$
$$\left. -r_\theta(\xi_{T-\lfloor\tau\rfloor}(w_{\lfloor\tau\rfloor}, y), T-\lfloor\tau\rfloor; \mu)\right\}$$
$$= \lambda(T-\lfloor\tau\rfloor)\left\{\tilde{s}(m_{T-\lfloor\tau\rfloor}(w_{\lfloor\tau\rfloor}, y), T-\lfloor\tau\rfloor; \mu) + m_{T-\lfloor\tau\rfloor}(w_{\lfloor\tau\rfloor}, y)\right.$$
$$\left. -r_\theta(R_{T-\lfloor\tau\rfloor}^{-1}m_{T-\lfloor\tau\rfloor}(w_{\lfloor\tau\rfloor}, y), T-\lfloor\tau\rfloor; \mu)\right\}$$

and now it follows by reversing time and applying Lemma B.4 that

$$\mathbb{E}\int_0^{T-\delta}\|\mathcal{I}_2(\tau)\|_H^2\, d\tau = \varepsilon_{LOSS},$$

where $\varepsilon_{LOSS}$ is given by Equation 20.

**Contribution from $\mathcal{I}_3$:** By triangle inequality and the assumption on uniform Lipschitzness of $r_\theta$ we have

$$\left\|\mathcal{I}_3(\tau)\right\|_H \le \lambda(T-\lfloor\tau\rfloor)L_s(T-\lfloor\tau\rfloor)\left\|\xi_{T-\lfloor\tau\rfloor}(w_{\lfloor\tau\rfloor}, y) - \xi_{T-\lfloor\tau\rfloor}(v_{\lfloor\tau\rfloor}, y)\right\|_H$$
$$+ \lambda(T-\lfloor\tau\rfloor)e^{(T-\lfloor\tau\rfloor)/2} + 1)\left\|w_{\lfloor\tau\rfloor} - v_{\lfloor\tau\rfloor}\right\|_H$$
$$\le \kappa_s(T-\lfloor\tau\rfloor)\left\|w_{\lfloor\tau\rfloor} - v_{\lfloor\tau\rfloor}\right\|_H,$$

where we abbreviate $\kappa_s(\tau') = L_s(\tau')\lambda(\tau')^2 + \lambda(\tau')e^{\tau'/2} + 1$ for convenience. Note that by assumption $\kappa_s(\cdot) \in L^2[\delta, T]$.

Combining the estimates, we obtain

$$\mathbb{E}\left\|w_{T-\delta} - v_{T-\delta}\right\|_H^2 \lesssim \varepsilon_{NUM} + \varepsilon_{LOSS} + \varepsilon_{INIT} + \mathbb{E}\int_0^{T-\delta}\kappa_s(T-\lfloor\tau\rfloor)^2\left\|w_{\lfloor\tau\rfloor} - v_{\lfloor\tau\rfloor}\right\|_H^2\, d\tau.$$

Applying Grönwall's inequality, it follows

$$\mathbb{E}\left\|w_{T-\delta} - v_{T-\delta}\right\|_H^2 \lesssim (\varepsilon_{NUM} + \varepsilon_{LOSS} + \varepsilon_{INIT})\exp\left(\int_0^{T-\delta}\kappa_s(T-\lfloor\tau\rfloor)^2 d\tau\right).$$

We may factor in the truncation by utilizing Equation 37

$$\mathbb{E}\|w_T - w_{T-\delta}\|_H^2 = \mathbb{E}_{y\sim\pi_y}\mathbb{E}_{x\sim\mu^y, z\sim\mathcal{N}(0,\mathcal{C})}\left\|(1-e^{-\delta/2})x + \sqrt{1-e^{-\delta}}z\right\|_H^2$$
$$\lesssim (1-e^{-\delta/2})^2\mathbb{E}_{x\sim\mu}\|x\|_H^2 + (1-e^{-\delta})\mathrm{tr}_H(\mathcal{C})$$
$$= \mathcal{O}(\delta).$$

Hence, we conclude

$$\mathbb{E}\left\|w_T - v_{T-\delta}\right\|_H^2 \lesssim (\varepsilon_{NUM} + \varepsilon_{LOSS} + \varepsilon_{INIT}) \exp\left(2\int_0^{T-\delta} \kappa_s(T - \lfloor\tau\rfloor)^2 d\tau\right) + \delta,$$

which yields the result. □

## D Details of numerical implementation

**Neural network architecture** In Figure 7, we highlight the FNO architecture that we use to approximate the score function in our approach. Here, we implement 4 hidden layers and $s$ denotes the number of pixels in both horizontal and vertical directions, while $h$ represents the number of hidden nodes. The Conditional method employs the same architecture but omits the transform $f(t)$ and includes an additional input node $y$. This modification affects only the dimensionality of the first layer, changing it to $\mathbb{R}^{s\times s\times 3}$, while all subsequent layers remain unchanged. Consequently, the total number of parameters remains in the same order. The unconditional score approximation also follows the same architecture as UCoS but without the transform $f(t)$.

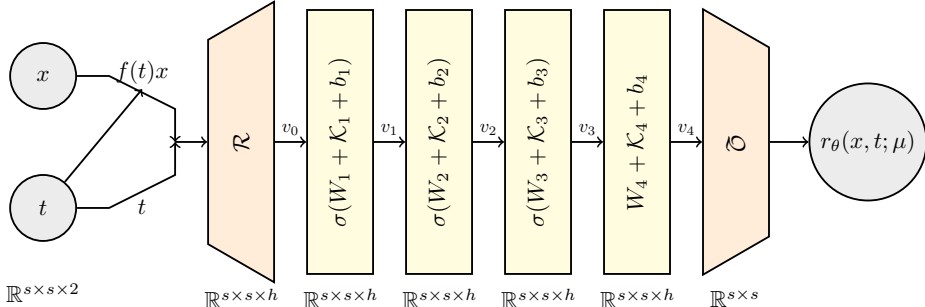

Figure 7: Our FNO architecture. The scalar multiplication is given by $f(t)x = \frac{x}{1+\text{std}(x)}$. Activation function $\sigma$ is given by Relu. The lifting block $\mathcal{R}$ is given by a linear layer and $\mathcal{Q}$ by two linear layers and an activation function. The Fourier layers are followed by a batch–normalization layer before the activation function. Below each layer is the dimensionality of a tensor after passing through each layer

We use the FNO architecture from Baldassari et al. (2024a) and choose $\delta = 5 \cdot 10^{-3}$ and $T = 1$. Similar to the works of Pidstrigach et al. (2024); Baldassari et al. (2024a), we run the forward SDE with a non-constant speed function leading to the SDE

$$dX_t = -\frac{1}{2}\beta(t)X_t dt + \sqrt{\beta(t)\mathcal{C}}dW_t$$

with $\beta(t) = 0.05 + t(10 - 0.05)$.

We train the neural network for 10 epochs for the 32-nodes architecture, 30 epochs for the 64-nodes architecture and 65 epoch for the 128-nodes architecture, where training is done on a Nvidia A100 GPU with 80 GB of memory. Training takes from 14-46 minutes for the unconditional method (depending on the architecture), 17-47 minutes for the conditional method and 24-51 minutes for UCoS.

During the training process we use the Adam stochastic optimizer with linearly decaying learning rate ranging from 0.002 to 0.0005. Samples are generated on the same machine. A uniform Euler–Maruyama approximation with 1000 steps is employed. Additional information can be found in the numerical implementation `https://github.com/FabianSBD/SBD-task-dependent/tree/UCoS`.

**Comparison methods** For clarity, let us give the precise score approximation of the following methods that we compare our method with. Note that some of these methods are used in combination with some other sampling method in the reference but we will always utilise backwards in time Euler–Maruyama approximations.

- SDE ALD uses the score approximation of Jalal et al. (2021), which is given by

$$s_\theta(x, t; \mu^y) = s(x, t, \mu) + A^*(\Gamma + \gamma_t I)^{-1}(y - Ax) \tag{38}$$

  for the hyper-parameter $\gamma_t$ under the assumption $\Gamma = \sigma^2 I$ for some $\sigma^2 > 0$. In line with Feng et al. (2023), we tune $\gamma_t$ such that the additive term has equal norm to the score function.

- DPS Chung et al. (2023a) employs a similar idea to SDE ALD by removing the hyper–parameter $\gamma_t$ and changing the mean of the Gaussian likelihood to be an estimate of $x_0$:

$$s_\theta(x, t; \mu^y) = s(x, t, \mu) - \rho \nabla_x \|y - A(\hat{x}_0(x))\|_2^2$$

  for some $\rho$ which is chosen such that $\rho = \xi / \|y - A(\hat{x}_0(x))\|$ for some constant $\xi$. We use a grid-search algorithm to find the optimal $\xi$. Above $\hat{x}_0(x)$ is an estimate of $\mathbb{E}(X_0|X_t = x)$ using the definition of the score function Equation 2.1.

- Proj Dey et al. (2024): This projection-based approach adds a data consistency step before every reverse time Euler–Maruyama step:

$$x_t \;=\; (\lambda A^\top A + (1 - \lambda)I)^{-1} \left((1 - \lambda)x_t' + \lambda A^\top y_t\right).$$

  We tune the hyper-parameter $\lambda$ by a grid search algorithm. Since the operator $(\lambda A^\top A + (1 - \lambda)I)^{-1}$ is very large in terms of GPU-memory, we employ an iterative scheme with 10 iterations at each time step to solve the inverse (CT imaging problem) or a very small batch-size (Deblurring problem). This significantly increases runtime compared to applying a precomputed operator.

- Conditional Baldassari et al. (2024a): This approach approximates the conditional score function

$$s(x, t; \mu^y) = -\frac{1}{1 - e^{-t}} \left(x - e^{t/2}\mathbb{E}(X_0^\mu|X_t^\mu = x, Y = y)\right)$$

  directly.

# E   Additional figures

More samples and in higher resolution for all methods can be found below in Figure 8 for the inpainting problem, in Figures 9, 10 and 11 for the CT imaging problem and Figure 12 for the Deblurring problem.

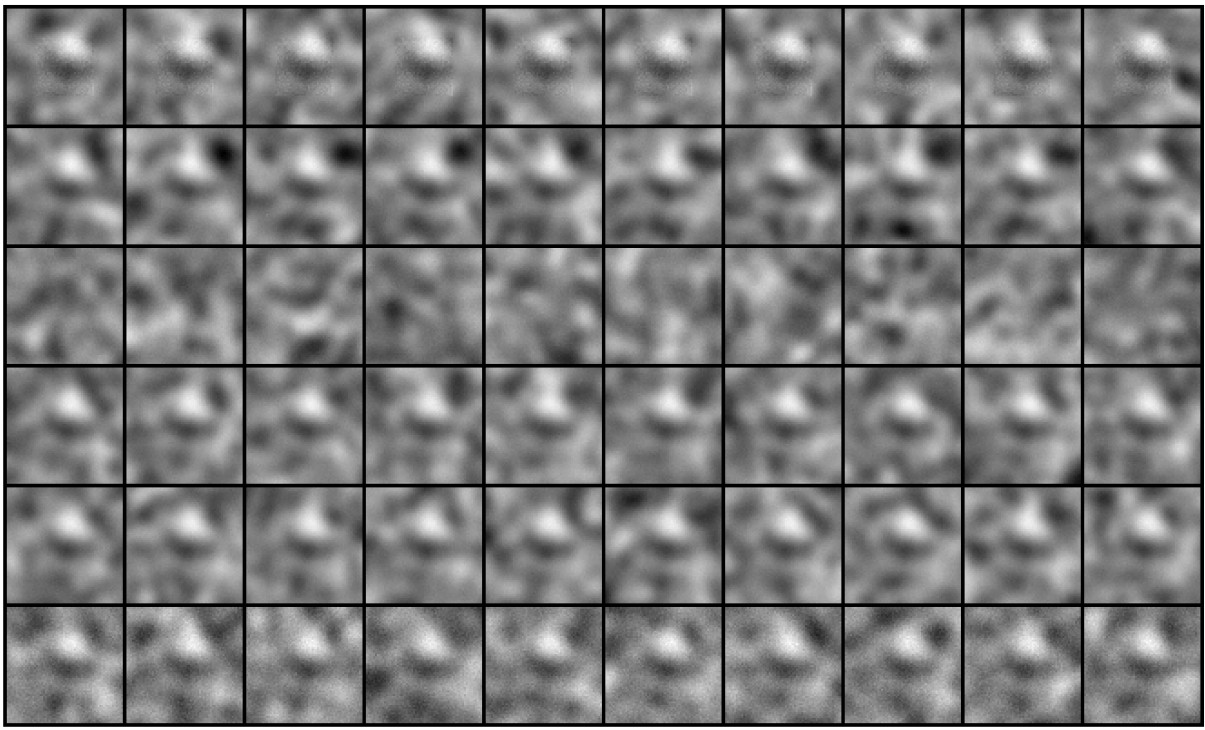

Figure 8: Posterior samples for Inpainting problem. Methods from top to bottom: SDE ALD, DPS, Proj, Conditional, UCoS, true posterior

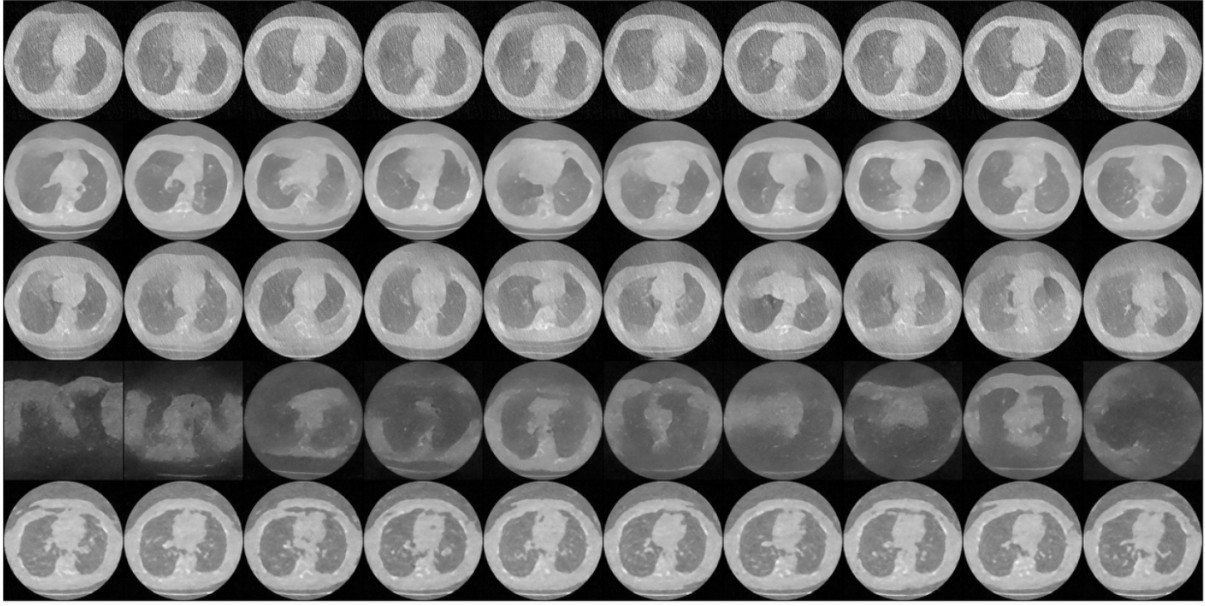

Figure 9: Posterior samples for CT imaging problem with a FNO architecture that uses 32 nodes per layer. Methods from top to bottom: SDE ALD, DPS, Proj, Conditional, UCoS.

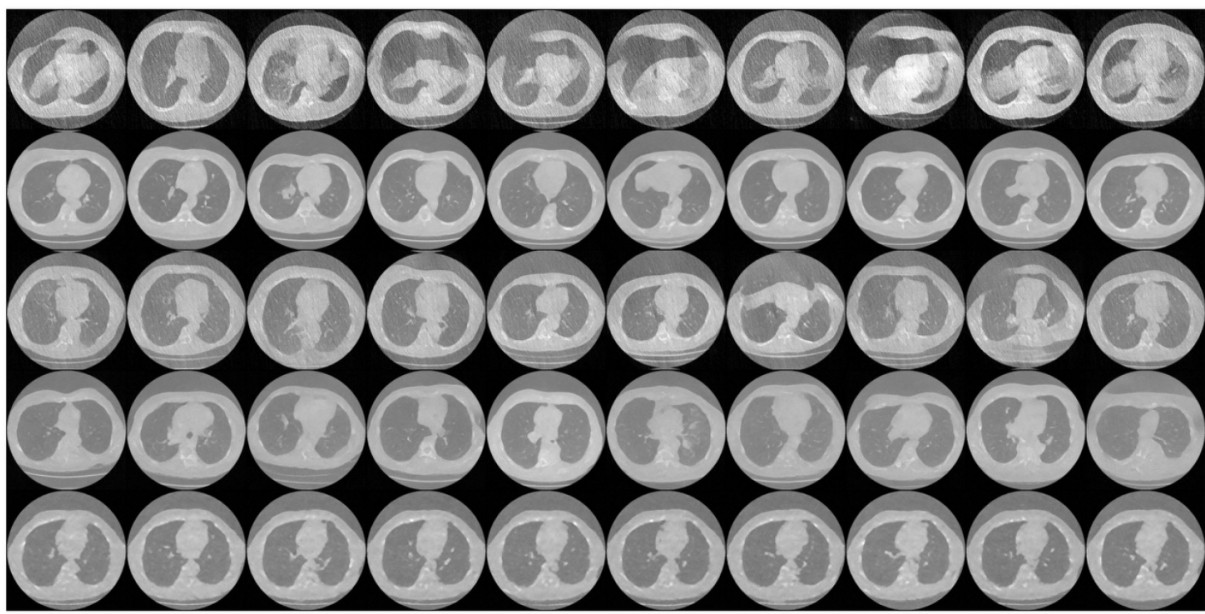

Figure 10: Posterior samples for CT imaging problem with a FNO architecture that uses 64 nodes per layer. Methods from top to bottom: SDE ALD, DPS, Proj, Conditional, UCoS.

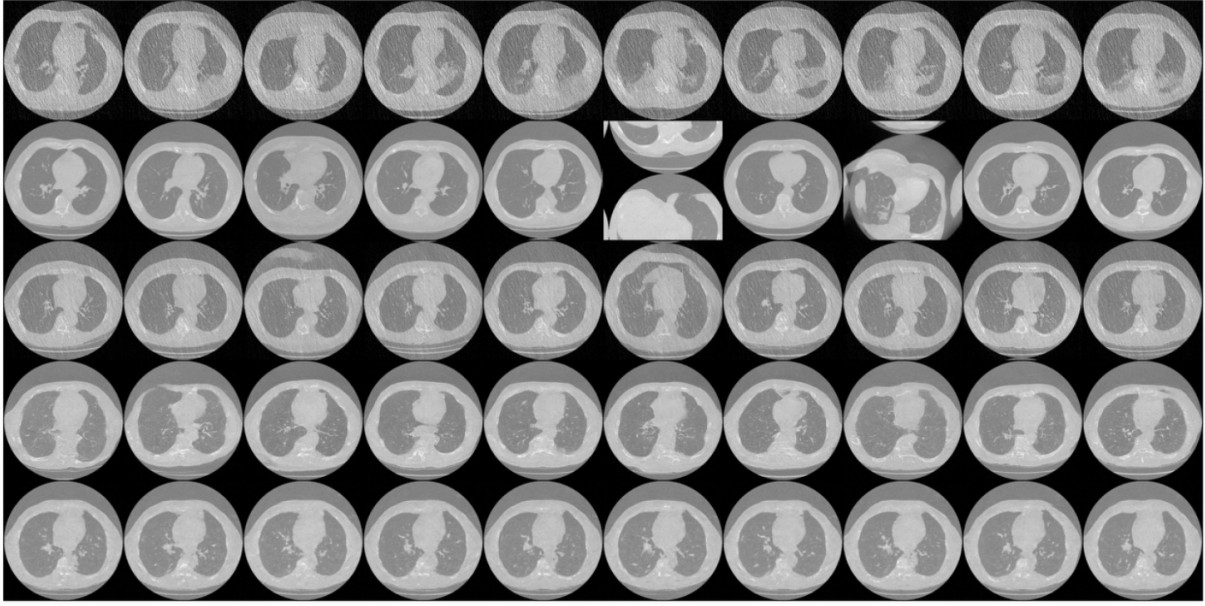

Figure 11: Posterior samples for CT imaging problem with a FNO architecture that uses 128 nodes per layer. Methods from top to bottom: SDE ALD, DPS, Proj, Conditional, UCoS.

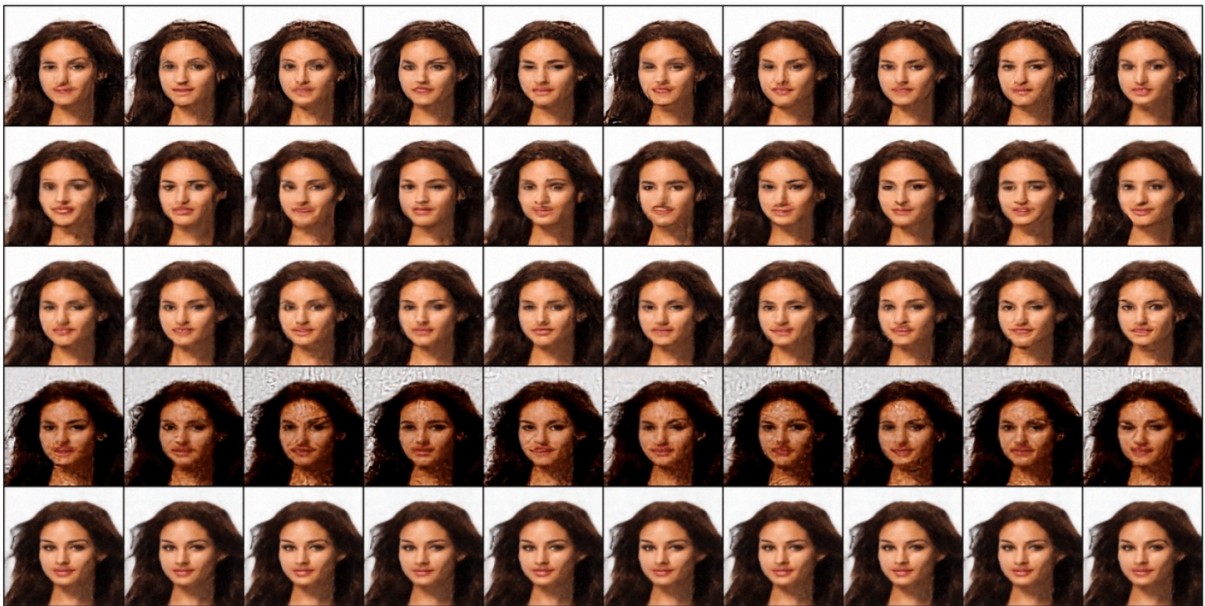

Figure 12: Posterior samples for Deblurring problem. Methods from top to bottom: SDE ALD, DPS, Proj, Conditional, UCoS.

Below in Figure 13, we reprint Figure 5 in larger size.

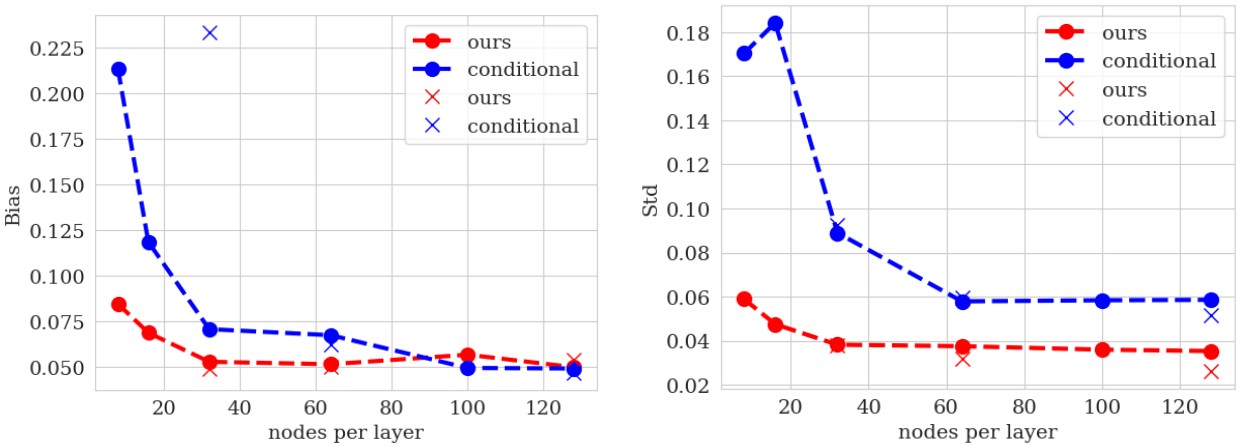

Figure 13: Comparison of the error dependence of the conditional method and UCoS on the parametrization of the score approximation. X-axis: number of nodes per layer in the FNO, Y-axis: L2 norm of the bias (left) or std (right), averaged over the number of pixels. Dashed lines correspond to $64 \times 64$ resolution and crosses correspond to the problem with $256 \times 256$ resolution.

