# OpenReview forum: "An Unconditional Representation of the Conditional Score in Infinite Dimensional Linear Inverse Problems"
_TMLR — Accepted by TMLR_

### Review · Reviewer_pG4G · 2025-07-24

**Summary Of Contributions:**

This work covers Bayesian inference in infinite dimensions using score-based diffusion models. Specifically, an "unconditional representation of the conditional score (UCoS)" is introduced which lowers computational costs as forward model evaluations are avoided during sampling. The main contribution of this work is theoretical, i.e., in the proofs of (1) that the conditional score is connected to a task-dependent unconditional score which can be learned in an offline phase, and (2) the convergence results that quantify how far the UCoS samples are from the true posterior.

**Audience:**

Yes

**Broader Impact Concerns:**

None.

**Claims And Evidence:**

No

**Requested Changes:**

I have some questions  that are crucial for my understanding of this work and I want the authors to reply to here and in a revision.

1. Why is it beneficial that *"the computational effort can be shifted to the offline task of training the unconditional score of a specific diffusion-like random process*". Relatedly, is it really a shift of computational effort, or does its amount change (decrease or increase)?
2. What is the practical benefit of the infinite dimensional setting (if there is any)? Does it allow to handle new types of tasks or data; is it more natural/appropriate than the finite dimensional setting?
3. Your submission largely builds upon the work by Baldassari et al., (2024a). In which terms is is similar, and where precisely does it differ. I have seen the comment *"This is in stark contrast to the conditional
approach Baldassari et al. (2024a), where the conditional score $(x, y, t) \mapsto s(x, t; \mu^y ) : R^n \times R^m \times R → R^n$ in Def. 2.3 is approximated"*, but it was too brief and superficial to answer my questions.

4. Minor. You state that *"This method extends rigorously to infinite-dimensional diffusion models, making it attractive for large-scale inverse problems, as it eliminates the need to evaluate the forward map during sampling."* How is the causality here? Do you mean that the method extends to infinite dimensions because there is no need to evaluate the forward map during sampling. If so, why?

**Strengths And Weaknesses:**

Regrettably, I am not familiar enough with Bayesian inference in infinite dimensions or using diffusion models to adequately assess the soundness of the theoretical part of this submission, which is a pity, as it is the main contribution. Therefore, this review is with low confidence.
That being said, as far as I can tell, a major strength of this work is its technical rigor. The proposed method is grounded in theoretical arguments and insights and comes with mathematical guarantees.
However (mostly unrelated to the acceptance criteria), with my level of expertise I found the submission quite unaccessible and I would have appreciated more discussions and interpretations of the results, e.g. in form of a short paragraph in the vincinity of the theorems and throughout.

The empirical evaluation (which is clearly not the focus of this work) is rather limited and from my perspective serves only as a proof of concept. Only two tasks (CT-Imaging and Deblurring) are studied and each only on the basis of a single dataset.
More importantly, **the experiments do not verify the central claim of the submission**, i.e., that "[the authors] address the prohibitive sampling cost for large-scale inverse problems by demonstrating that this computational overhead can be offloaded to the training phase, thereby accelerating posterior sampling." Looking at the reported runtime, the method is exactly as fast as its most related baseline "Conditional" (Baldassari et al., 2024a).   Therefore, I currently answered no wrt claims and evidence.

---

> ### Author Response · Authors · 2025-09-13
> **Response to Reviewer pG4G**
>
> We thank the reviewer for their time and provided feedback that highlights how we can improve our work.
>
>
> We agree to change the wording of the claim in question ("We address the prohibitive sampling cost for large-scale inverse problems by demonstrating that this computational overhead can be offloaded to the training phase, thereby accelerating posterior sampling") to reflect that the computational advantage against conditional method is in a lower dimensional function approximation.
>
> Let us address the mentioned questions next. We agree that these are important points to understanding this approach and we will be sure to make these clearer in a revision.
>
> 1. We are indeed shifting computational effort to training from sampling phase and the motivation for this is an imaging task with fixed geometry (consider medical imaging) repeated over many different measurement data (many patients): For the CT problem and 32 node architecture our method needs 24 min / epoch and unconditional needs 14 min / epoch. During sampling we need 15 mins / 1000 samples and unconditional methods need from 35 to 224 mins / 1000 samples. Consequently, shifting computational benefit to the training phase results in lower total computational time for training and sampling e.g. if 1000 samples are generated for 10 or more imaging problems.
>
> 2. Beyond the theoretical advantage of formulating the inverse problem in the undiscretized setting that reflects the physical model, the practical benefit is that our results hold independently of resolution.
>     In contrast, several error bounds in the literature depend explicitly on the dimension of the unknown (see, e.g., [1, 2]). For high-dimensional problems these bounds can include prohibitively large constants, limiting their practical utility. Our results avoid this issue, since they remain valid for any discretization.
>
> 3. For a detailed explaination of the similarities and differences between UCoS and Conditional, we refer to the Central response.
>
> 4. To clarify, our method does not extend to infinite dimensions because forward evaluations are avoided. Instead, the infinite-dimensional setting is the natural framework for Bayesian inverse problems, where discretization-invariance is a desirable property in both theory and practice. Within this framework we derive the affine identity for the conditional score, and as a result forward evaluations are not needed at inference. In large-scale inverse problems, parameters to be inferred are often of very high or infinite dimension, which usually leads to expensive evaluations of the forward map. Since our method does not require this computation in sampling, it can handle large-scale inverse problems in this regard.
>
> **References**
>
> [1] Joe Benton et al. Nearly d-Linear Convergence Bounds for Diffusion Models via Stochastic
> Localization. 2024. arXiv: 2308.03686 [stat.ML]. url: https://arxiv.org/abs/2308.03686.
>
> [2] Sitan Chen et al. Sampling is as easy as learning the score: theory for diffusion models with
> minimal data assumptions. 2023. arXiv: 2209.11215 [cs.LG]. url: https://arxiv.org/abs/2209.11215.

---

### Review · Reviewer_9hUC · 2025-08-29

**Summary Of Contributions:**

The paper proves that, in linear–Gaussian inverse problems, the posterior (conditional) score can be written as an affine transform of an unconditional score. Building on this, the authors propose training a task-dependent but unconditional diffusion model whose dynamics already encode the observation operator. With that model in hand, conditional sampling for the given inverse problem becomes a cheap affine post-processing step that avoids per-step applications of the forward operator.

# Questions
- “Matrix-free” meaning: Matrix-free seems to denote the fact that you do not need to evaluate or invert $A$ during sampling. This naming convention is a bit unusual to me, but maybe I missunderstood.

# Summary recommendation
A solid, well-argued paper with a useful and (to my knowledge) novel identity at its core. Recommend accept to TMLR, with the above additions likely to further improve clarity and empirical persuasiveness.

**Audience:**

Yes

**Broader Impact Concerns:**

The same as for any work in generative modelling.

**Claims And Evidence:**

Yes

**Requested Changes:**

# Changes (to strengthen the paper)
1) Running toy examples. Include worked-through cases (e.g., C = I, Gamma = epsilon * I, A a masking/projection operator) to give immediate intuition for the covariance shaping, the R_t / Sigma_t constructions, and the resulting per-coordinate noise levels.
2) Direct use of the conditional-score identity. For a few problems where A is cheap (e.g., inpainting/masking) or moderately sized, demonstrate literal use of the affine conditional-score formula (even if slower) to show the identity’s practical fidelity and to triangulate performance against the proposed task-dependent training. Here, comparison with DPS would also be of interest to the community, even if the results are negative.

**Strengths And Weaknesses:**

# Strengths
- The main identity (Eq. 15) appears novel and of independent interest: it cleanly reduces a conditional score to an affine transform of a task-dependent unconditional score.
- It outperforms the “Conditional” baseline (a model trained directly on (x, y) to approximate the conditional score) in several benchmarks.
- The exposition is clear and the paper is well structured.

# Weaknesses
- The approach is not training-free: a new task-specific unconditional model must be trained for each linear inverse problem. In terms of wall-clock, the reported training and evaluation times are broadly comparable to directly training a conditional model for that task.
- While improvements over the conditional baseline are present, the gains are within the reported standard deviations, so some margins appear modest.

---

> ### Author Response · Authors · 2025-09-13
> **Response to Reviewer 9hUC**
>
> We thank the reviewer for their thorough analysis and feedback.
>
> Quoted: *''Matrix-free'' meaning: Matrix-free seems to denote the fact that you do not need to evaluate or invert during sampling. This naming convention is a bit unusual to me, but maybe I missunderstood.*
>
> We say that a method is matrix-based if we need to explicitly construct a matrix representation of the forward operator $A$. In practice this would be done by setting $A_{i, :} := A(e_j)$ for the canonical basis $(e_j)_{j = 1}^n$, where all coordinates are zero, except for $j$. Computing this matrix requires $n$ evaluations of the forward map and saving it requires memory of order $m n$. In the CT example and on our setup, computing this matrix takes around one hour and saving it requires around 32 GB of GPU memory.
>
> In contrast, a matrix-free method requires only evaluation of the forward and adjoint maps $A(x), A^*(y)$ given by an expert for any $x, y$ and there is no necessity to compute and save the corresponding matrix.
>
>
> Quoted:*The approach is not training-free: a new task-specific unconditional model must be trained for each linear inverse problem. In terms of wall-clock, the reported training and evaluation times are broadly comparable to directly training a conditional model for that task. While improvements over the conditional baseline are present, the gains are within the reported standard deviations, so some margins appear modest.*
>
> Both UCoS and Conditional indeed require task-specific training, which is a trade-off for avoiding forward map evaluations, which we offer as an alternative to existing unconditional approaches. To compare directly with Conditional, our results show a qualitative difference in how the two methods perform depending on network size. In Figure 3, UCoS consistently outperforms Conditional in terms of both bias and standard deviation when using lighter architectures. This demonstrates that UCoS achieves good posterior quality with significantly smaller networks, reflecting the reduced complexity of the unconditional score formulation. For larger architectures, Conditional does improve and in some cases approaches UCoS in terms of bias, which we attribute to the manifold hypothesis, that is, the joint distribution of $(x,y)$ may well be supported on a structured lower-dimensional manifold rather than the full product space.
>
> **Requested Changes**
> Quoted: *Running toy examples. Include worked-through cases (e.g., C = I, Gamma = epsilon * I, A a masking/projection operator) to give immediate intuition for the covariance shaping, the $R_t / Sigma_t$ constructions, and the resulting per-coordinate noise levels.
> Direct use of the conditional-score identity. For a few problems where A is cheap (e.g., inpainting/masking) or moderately sized, demonstrate literal use of the affine conditional-score formula (even if slower) to show the identity’s practical fidelity and to triangulate performance against the proposed task-dependent training. Here, comparison with DPS would also be of interest to the community, even if the results are negative.*
>
> We thank the reviewer for this suggestion. We provide a visualization of the per-coordinate noise levels of $\Sigma_t$ for a masking operator and the CT--problem.
> Further for an inpainting problem and with Gaussian prior, we explicitly solve the score functions to demonstrate the fidelity of our approach and conditional.
>
> We have attached these toy examples under the following link.
> https://drive.google.com/file/d/1K45vwrJ3ITQjbFdkR_xD7HEJPBnm-14q/view

---

### Review · Reviewer_SMGZ · 2025-09-04

**Summary Of Contributions:**

The paper addresses the problem of posterior sampling in Bayesian linear inverse problems with additive Gaussian noise, where the posterior $p(x \mid y)$ is typically intractable. Score-based diffusion models (SDMs) have recently been proposed to approximate such posteriors, but existing methods often require repeated evaluation of the forward operator $A$ and its adjoint $A^*$ during inference, which becomes computationally expensive in high-dimensional or infinite-dimensional settings.

To address this, the authors propose **UCoS**, an unconditional representation of the conditional score function. The central idea is to shift the computational cost to an offline training phase, where a task-dependent but measurement-independent denoising network is trained. At inference, the conditional score is recovered by applying an affine transformation to the latent variable and measurement, without requiring further applications of $A$ or $A^*$.

The method is formulated in infinite-dimensional Hilbert spaces to ensure discretization-invariance and is supported by a theoretical identity connecting the conditional score to the trained unconditional score, as well as a convergence result bounding the sampling error.

The authors validate their approach on two imaging inverse problems: *computed tomography (CT) reconstruction* using the LIDC-IDRI dataset and *image deblurring* using the CelebA dataset. They compare UCoS with different baselines, including conditional score-based diffusion, DPS, Proj, and SDE ALD. The reported results show that UCoS achieves posterior samples of comparable or superior quality to these baselines, while maintaining similar runtime despite not applying $A$ during inference.

The authors conclude that UCoS provides a theoretically principled and practically effective framework for posterior sampling in high-dimensional inverse problems, with the key advantage of decoupling inference from forward operator evaluations.

**Audience:**

Yes

**Claims And Evidence:**

No

**Requested Changes:**

- Provide a theoretical comparison of the per-step computational complexity of UCoS versus the Conditional baseline, explicitly comparing the cost of applying $A x$ versus evaluating $r_\theta(\xi(x, y), t)$.

- Add an ablation study to analyze the trade-off between network complexity, posterior sample quality, and runtime. In particular, test whether a lighter network architecture could offer improved inference time without degrading performance.

- Clarify and moderate the paper’s claims regarding inference efficiency, especially in light of the empirical runtimes being similar to baselines that do apply $A$.

- Include additional experiments (e.g., with a more complex forward model or simpler network) that clearly demonstrate the benefit of decoupling from operator applications during inference.

- Improve the structure of the paper by separating high-level explanations from detailed derivations. Consider moving some theoretical sections (e.g., Section 3.1) to the supplementary material.

- Revise figure annotations to include model and axis labels directly within the figures. Additionally, update figure captions to briefly summarize the key takeaway from each figure or table.

- Correct minor typographical issues, such as repeated words (e.g., “equation equation”).

**Strengths And Weaknesses:**

**Strengths.**
The central idea of the paper is interesting and mathematically well grounded. The authors provide a precise formulation in infinite-dimensional Hilbert spaces, supported by a clear theoretical identity and convergence guarantees. This is a strong point of the work.

---

**Weaknesses.**
There are concerns regarding the practicality of the proposed framework, particularly in terms of the computational complexity of the final model and its implementation. The paper does not provide an order-of-complexity analysis of inference, which makes it difficult to assess how the theoretical advantages translate into practice. Moreover, the experimental results, while demonstrating that UCoS achieves comparable posterior quality to existing baselines, do not convincingly support the central claim of improved runtime efficiency. In below, I will explain my concerns in detail.

**Major comment on the clarification of the runtime efficiency claims:**

The authors claim that UCoS achieves more efficient inference by avoiding applications of the forward operator $A$ during sampling. However, the empirical runtimes reported in Tables 1 and 2 are identical to those of the Conditional baseline, which does apply $A$ at inference. This suggests that the claimed computational savings are not realized in practice. One possible reason is that the UCoS framework still requires evaluating a neural network $r_\theta(\xi(x, y), t)$ at every sampling step. Therefore, although $A$ is not directly applied to $x$ at inference, the network $r_\theta$ still processes a full $x$-dependent input and performs a costly forward pass, particularly given the use of a Fourier Neural Operator (FNO) as the backbone.

Therefore, to support the central hypothesis that eliminating applications of $A$ leads to more efficient inference, the authors should provide a clear theoretical comparison of the per-step computational complexity between UCoS and Conditional. In particular, they should quantify the cost of applying $A x$ versus evaluating $r_\theta(\xi(x, y), t)$. Furthermore, an ablation study is necessary to assess the trade-off between network complexity, denoising performance, and runtime. For instance, it is unclear whether simpler architectures could retain similar posterior quality while improving inference time, or whether the cost of evaluating $r_\theta$ outweighs the benefit of removing applications of $A$. None of these analyses are presented in the current version, and as a result, the paper does not convincingly support its core claim regarding the efficiency of the proposed formulation, neither theoretically nor experimentally.

In fact, although the authors acknowledge the similar complexity of the FNO backbone used in both methods, this undermines the central motivation for the UCoS reparameterization. If the neural network architecture ultimately dominates inference time, then moving operator applications offline does not yield tangible benefits in practice. This suggests that the complexity of posterior sampling, especially in high-dimensional or functional inverse problems, is not solely driven by the cost of operator evaluations, but also by the cost of score network evaluations and data flow. The authors are encouraged to clarify this point and moderate claims regarding efficiency gains, and provide additional experiments to demonstrate that decoupling from $A$ leads to clear speedups in a different regime (e.g., with a lighter network or a more complex forward model).

**Some minor comments:**

- The flow of the paper could be improved. At present, detailed derivations are somewhat merged with the high-level description of the approach. For clarity, I suggest moving parts of the theoretical development (e.g., Section 3.1 on theoretical foundations) to the supplementary material.

- Figures would benefit from clearer annotation. It would be preferable to indicate directly within the figure (not only in the caption) which model each row corresponds to and what each column represents. In addition, including a brief statement of the main conclusions drawn from the figures and tables in the captions would be helpful.

- There are minor typographical issues, for instance the word *equation* is occasionally repeated (e.g., “equation equation”).

---

> ### Author Response · Authors · 2025-09-13
> **Response to Reviewer SMGZ**
>
> We thank the reviewer for the detailed report and for recognizing the theoretical clarity and guarantees of our work. We respond to the main concerns below.
>
> **On weaknesses and major comment on the clarification of the runtime efficiency claims**
>
> Quoted: *The authors claim that UCoS achieves more efficient inference by avoiding applications of the forward operator A during sampling. However, the empirical runtimes reported in Tables 1 and 2 are identical to those of the Conditional baseline, which does apply A at inference. This suggests that the claimed computational savings are not realized in practice...*
>
> This is a misunderstanding: the Conditional method does *not* apply $A$ at inference (see our Central Response). Consequently, the main part of this comment does not apply.
> We also point out to the reviewer the scaling analysis at the end of Section 5, where we compare the two methods in terms of the complexity of the FNO backbone. It can be derived from here, for instance, with only 20 nodes per layer, UCoS archives approximately the same accuracy as 40 nodes per layer for Conditional (thus in this case, UCoS archives the same sample quality as Conditional for half the sampling time.) See also our additional figures in Appendix (especially Fig. 5 and Fig. 7, where Conditional basically fails to produce meaningful samples with the same sampling time as UCoS.)
>
>
> Quoted: *Therefore, to support the central hypothesis that eliminating applications of A leads to more efficient inference, the authors should provide a clear theoretical comparison of the per-step computational complexity between UCoS and Conditional. In particular, they should quantify the cost of applying A versus evaluating $r_\theta(\xi, y)$. Furthermore, an ablation study is necessary to assess the trade-off between network complexity, denoising performance, and runtime. For instance, it is unclear whether simpler architectures could retain similar posterior quality while improving inference time...*
>
>
> For both methods the cost of inference is one score network evaluation per step, but Conditional requires higher input dimensionality $(m+n+1)$ and empirically needs larger architectures to achieve accuracy comparable to UCoS, which translates into higher per-step complexity for Conditional. In particular, since UCoS and Conditional avoid applications of $A$ during inference, a per-step comparison between ''cost of applying $A$'' and ''cost of evaluating $r_\theta$'' is not applicable. See also our general response above.
>
> Quoted: *Furthermore, an ablation study is necessary to assess the trade-off between network complexity, denoising performance, and runtime. For instance, it is unclear whether simpler architectures could retain similar posterior quality while improving inference time...*
>
>
> Our scaling results in Section 5 and the Appendix already serve as such an ablation: UCoS achieves the same or better posterior quality with substantially lighter networks than Conditional (e.g., UCoS with 20 nodes matches Conditional with 40).
>
> **On minor comments** We thank the reviewer for the suggestions and for pointing out the typos. We will take these into account when preparing the revision.
>
> **On Requested Changes**
> Quoted: *Provide a theoretical comparison of the per-step computational complexity of UCoS versus the Conditional baseline, explicitly comparing the cost of applying $Ax$ versus evaluating $r_\theta(\xi(x, y), t)$.
> Add an ablation study to analyze the trade-off between network complexity, posterior sample quality, and runtime. In particular, test whether a lighter network architecture could offer improved inference time without degrading performance.*
>
> This is partly done in scaling analysis in Section 5, backed up by additional illustrative figures in Appendix -- we will expand the analysis to greater detail in the revision. Applying $Ax$ is not applicable.
>
> Quoted: *Clarify and moderate the paper’s claims regarding inference efficiency, especially in light of the empirical runtimes being similar to baselines that do apply A*
>
> To repeat ourselves, UCoS is much more efficient than other baseline methods that do apply $A$. For the Conditional method that does not apply $A$, UCoS outperforms Conditional with lighter network complexity.
>
> Other suggested changes are answered above.

---

> > ### Comment · Reviewer_SMGZ · 2025-10-03
> > **Response to the rebuttal**
> >
> > I thank the authors for the clarifications provided. It helped me better understand the distinction between the Conditional and Unconditional (UCoS) approaches. As the main text is theory-heavy, it was very difficult to extract and compare the core messages and contributions during the initial reads.
> >
> > As I mentioned previously, I find the central idea of the paper interesting and potentially impactful. However, I still believe a major revision is necessary for this paper to reach its potential. I itemize my opinions below:
> >
> > - **A clear, explicit comparison between Conditional and Unconditional methods** should be presented in terms of input dimensionality and resulting network complexity: i.e., *(m + n + 1)* (Conditional) vs. *(n + 1)* (UCoS). This directly supports the claim that UCoS requires lighter networks and should be emphasized in the main text.
> >
> > - **Figures 5 and 6 from the Appendix should be moved to the main text**, as they are central to validating the runtime and model complexity advantages of UCoS which are key contributions of the paper. Also, a clearer and more prominent explanation is required for these results.
> >
> > - **To improve the clarity and readability of the paper**, I strongly suggest moving much of the theoretical content to the supplementary material and instead focusing on the conceptual innovation and empirical justification. The current structure makes it difficult to stay focused on the core contributions, and it is easy to get lost in the extensive derivations.
> >
> > I also have major concerns about the experimental results, particularly regarding the qualitative evaluation:
> >
> > - **Comparing Figure 8 and Figure 9 at high resolution**, I find that the Conditional model with 128 nodes yields visually better samples than the Unconditional model with 32 or 64 nodes, even if the bias and std metrics appear similar. This raises questions about whether the chosen metrics sufficiently capture perceptual quality.
> >
> > - **Additionally, in some figures (e.g., Figure 9)**, the posterior samples generated by the Unconditional model appear unusually constant, with almost no variation except for the first two samples. This behavior seems unusual to me.
> >
> > - **Across several examples (e.g., Figure 1 and Figure 6)**, I find that DPS is qualitatively better compared to UCoS, particularly in terms of image sharpness and detail. This observation makes me doubt the performance of UCoS.

---

> > > ### Author Response · Authors · 2025-10-09
> > >
> > > We thank the reviewer for the follow-up comments. We will take your suggestions into account when preparing the revision, in particular, making sure that the core contributions are conveyed more clearly.
> > >
> > > Regarding the comments about the experimental results, we have the following remarks.
> > >
> > >  - We agree that quantitative metrics do not always fully capture perceptual or task-relevant quality, especially for medical images. For example, in Figure 9, sample 2 of Conditional method shows disconnected tissue and sample 8 has merged lung wings, both of which deviate anatomically from plausible lung structures. In contrast, all UCoS samples preserve realistic high-level shapes and structures.
> > >     We still believe that Figures 1, 5, and 6 (together with Figure 3) convincingly demonstrate that UCoS achieves performance saturation with fewer parameters and provides accurate posteriors faster, as also supported by Figure 2 and Table 2.
> > >
> > > - We assume that the reviewer refers to UCoS as the 'unconditional' method (note that we use the term 'unconditional methods' to denote other methods (SDE ALD, DPS and Proj) in the paper, due to the unconditional nature of their scores). We acknowledge that UCoS exhibits lower sample variance, which we view as a desirable property. Taking a closer look at Figures 7–9, UCoS samples are not constant, although they are indeed similar (and for good reason, as they resemble the ground truth).
> > >
> > > - This remark again highlights the central question of what is preferable in posterior samples. As shown in Figures 8 and 9, UCoS effectively captures the high-level structures of the ground truth, whereas DPS often introduces features that are not present in the true solution.
> > >     More importantly, DPS sampling is computationally expensive. As discussed in the introduction, evaluating the forward map multiple times during sampling becomes prohibitive for large-scale inverse problems, which is also reflected in the compute times reported in Tables 1–2 and in the tables accompanying Figures 5 and 6.
> > >     For instance, while the UCoS samples with 128 nodes in Figure 9 are, in our view, preferable to the DPS samples with 64 nodes in Figure 8, even though they are obtained at roughly half the computational cost.

---

### Author Response · Authors · 2025-09-13
**Central Response to all Reviewers**

We thank all reviewers for their constructive feedback. Before addressing specific comments, we briefly recall two main approaches to sampling for inverse problems:

- **Unconditional methods:** (DPS, Proj, SDE-ALD) sampling is learned independently of the inverse problem. Conditioning is then enforced during sampling, often approximately, by incorporating measurement information, which requires repeated forward operator evaluations.
- **Conditional methods:** (Baldassari et al.  (2024a)) sampling is learned directly from the distribution of $x$ given $y$. This requires modeling both the prior on $x$ and how measurements $y$ affect it, making the learning task higher-dimensional and more complex (see Figs.\ 2 and 5).

Neither our method (UCoS) nor the Conditional methods require forward operator evaluations at sampling time, unlike other tested methods (SDE-ALD, DPS, Proj). Therefore, comparison of UCoS to Unconditional methods is straightforward; in online sampling, UCoS evaluates an unconditional score of similar computational complexity as Unconditional methods. On top of that, Unconditional methods evaluate the forward mapping multiple times. Our key argument is that this computational overhead becomes prohibitive for large-scale models.
As this was not conveyed clearly enough, we will revise the paper to emphasize it.

Let us now focus on clarifying similarities and differences between UCoS and Conditional methods, as raised by all reviewers.
To enable a principled comparison between UCoS and conditional methods, one can try to (a) compare their runtimes when both are tuned to achieve roughly the same accuracy, or (b) compare their accuracies when both are trained with similar architectures that yield comparable runtimes. In this paper, we adopt approach (b), since - as we argue below - achieving similar accuracy with the conditional method requires a larger training dataset due to the higher input dimensionality of its score function, which would render the comparison unbalanced.

**Sample complexity**
The Conditional method must learn a score function $s(x,y,t): \mathbb{R}^n \times \mathbb{R}^m \times \mathbb{R} \to \mathbb{R}^n$, while UCoS requires only $s(x,t): \mathbb{R}^n \times \mathbb{R} \to \mathbb{R}^n$. The added dependence on $y$ increases the dimensionality of input space from $n+1$ to $n+m+1$, which substantially raises the required sample complexity and the optimal network size needed to achieve a given error tolerance.

For intuition, note that a Lipschitz function on a $d$-dimensional cube requires $\mathcal{O}(\epsilon^{-d})$ samples to reach uniform error $\epsilon$. In this toy case, the unconditional score would need $\epsilon^{-m}$ times fewer samples for same accuracy. Figure 3 illustrates this: UCoS attains good quality with smaller networks, while Conditional requires significantly larger models, especially at higher resolutions where UCoS saturates with better accuracy.

**Runtime comparisons**
We agree with reviewers that runtime analysis is subtle, since fair comparison requires controlling for network architecture. In the current version, we compared `wall-clock' runtimes of UCoS against other baselines (e.g., DPS, Proj, SDE-ALD). As pointed out above, for the Conditional method, we instead aim for the same network complexity (hence a similar runtime) to observe different quantitative performance. We will clarify this choice in the revision, and we agree that future experiments varying network architectures would further strengthen the runtime discussion. UCoS offers tangible benefits in regimes where forward operators are very costly or where smaller architectures suffice, which we will highlight more carefully in the revised text.

---

### Decision · Action_Editor_KhVG · 2025-10-21

**Recommendation:** Accept with minor revision

**Additional Comments:**

No consensus has been reached regarding this paper.

Indeed, the reviewers’ opinions were mixed. In particular:
- The presentation of the paper should be improved, following the reviewers’ feedback.
- Some of the claims should be revised, as the supporting arguments are unconvincing.

Despite these criticisms, I have decided to recommend acceptance, on the condition that the authors revise their manuscript accordingly.
In particular, I ask the authors to:

- Include the material they promised to add in their responses to the reviewers.
- Revise the presentation of the method to improve clarity and restructure the exposition around the core contributions. To this end, I suggest that the authors first present their contribution in finite dimensions. In my opinion, the key ideas underlying all derivations in the paper are that (1) the target distribution of the Ornstein–Uhlenbeck process is not a standard Gaussian but a Gaussian with a general covariance matrix, and (2) the authors employ a preconditioned version of the OU diffusion. The extension to the infinite-dimensional case is certainly interesting, but it may appear mainly technical to some readers.
- Moderate their claims that the method “accelerates posterior sampling” or is “superior". Indeed, standard posterior sampling does not require retraining, whereas in this work, posterior sampling is achieved by modifying the training of the prior. This is a fundamentally different approach. In my view, this distinction is precisely the main contribution of the paper.
- Acknowledge that the experimental results are relatively inconclusive and that further work is needed to fully demonstrate the benefits of the proposed method.

**Audience:**

Yes

**Audience Explanation:**

The derivation of the methodology proposed in the paper, for designing infinite-dimensional prior distributions and sampling the resulting posterior distributions arising from linear inverse problems, is worthy of publication and should be of interest to the community.

**Claims And Evidence:**

Yes

**Claims Explanation:**

This paper addresses the problem of posterior sampling in infinite-dimensional settings. More precisely, the goal is to sample from a posterior distribution associated with a linear inverse problem and a learned prior.

The first contribution of the paper is to show that the family of scores (or denoisers) associated with the posterior can be directly related to the quantities arising from the inverse problem and the family of scores (or denoisers) corresponding to the prior. Based on this observation, the authors propose a new parametrization of these functions, which can be trained without requiring access to the observation of the signal to be reconstructed.

Using arguments similar to those in existing literature, the paper derives theoretical guarantees for the resulting generative models under appropriate conditions.

Finally, the paper concludes with modest experimental results that serve as a proof of concept.

The technical derivations appear to be correct; however, some of the claims strike me as misleading or partially inaccurate. In addition, the experimental evaluation is rather limited.

**Resubmission Of Major Revision:**

The authors may consider submitting a major revision at a later time.